# Clinically Expired Platelet Concentrates as a Source of Extracellular Vesicles for Targeted Anti-Cancer Drug Delivery

**DOI:** 10.3390/pharmaceutics15030953

**Published:** 2023-03-15

**Authors:** Ana Meliciano, Daniela Salvador, Pedro Mendonça, Ana Filipa Louro, Margarida Serra

**Affiliations:** 1iBET, Instituto de Biologia Experimental e Tecnológica, Apartado 12, 2781-901 Oeiras, Portugal; 2ITQB-NOVA, Instituto de Tecnologia Química e Biológica António Xavier, Universidade Nova de Lisboa, Avenida da República, 2780-157 Oeiras, Portugal; 3IPST, Instituto Português do Sangue e da Transplantação, 1740-005 Lisboa, Portugal

**Keywords:** expired platelet concentrates, platelet-derived extracellular vesicles, drug delivery system, paclitaxel, anti-angiogenic potential, human breast cancer cell line

## Abstract

The short shelf life of platelet concentrates (PC) of up to 5–7 days leads to higher wastage due to expiry. To address this massive financial burden on the healthcare system, alternative applications for expired PC have emerged in recent years. Engineered nanocarriers functionalized with platelet membranes have shown excellent targeting abilities for tumor cells owing to their platelet membrane proteins. Nevertheless, synthetic drug delivery strategies have significant drawbacks that platelet-derived extracellular vesicles (pEV) can overcome. We investigated, for the first time, the use of pEV as a carrier of the anti-breast cancer drug paclitaxel, considering it as an appealing alternative to improve the therapeutic potential of expired PC. The pEV released during PC storage showed a typical EV size distribution profile (100–300 nm) with a cup-shaped morphology. Paclitaxel-loaded pEV showed significant anti-cancer effects in vitro, as demonstrated by their anti-migratory (>30%), anti-angiogenic (>30%), and anti-invasive (>70%) properties in distinct cells found in the breast tumor microenvironment. We provide evidence for a novel application for expired PC by suggesting that the field of tumor treatment research may be broadened by the use of natural carriers.

## 1. Introduction

Platelet concentrates (PC) are blood-derived products enriched in functionally active platelets from healthy donors and have become one of the most effective and vital medical products, as evidenced by their inclusion in the World Health Organization “List of Essential Medicines” [1]. Platelet transfusion therapy is widely used in clinical practice for acute hemorrhage treatment (therapeutic transfusion) or preventively (prophylactic transfusion) in patients with thrombocytopenia or severe platelet function abnormalities [2].

Over the past few years, there has been an increased demand for platelet components, and a similar trend is expected in the upcoming years [3]. Aside from the increasing incidence and prevalence of hematological malignancies and an increasingly aging population, the limited shelf life of PC could also be a factor underlying the rise in PC demand [3,4]. Indeed, PC for transfusion have a shelf-life of 5–7 days from the date of collection, as extended storage periods have an increased risk of microbial contamination [4]. Due to this limitation, along with the requirement to maintain an adequate inventory of PC to fulfill emergency transfusion needs, 10–20% of platelet donations are discarded owing to expiration [4].

To address PC wastage, advances towards improving PC utilization practices and alternative applications for outdated PC have been studied. Among these, the use of PC as a cell culture supplement and therapeutic product for regenerative medicine has been extensively reported because of the growth factor (GF) cargo enclosed in platelet α-granules and released during PC storage [5]. Indeed, our group and others have shown that human platelet lysates prepared from expired human PC are a feasible and safe xeno-free alternative substitute for gold standard fetal bovine serum (FBS) supplementation for ex vivo expansion of mesenchymal stem cell cultures [6,7], human dermal fibroblasts [8], and adipose-derived stromal cells [9], among other primary cells.

A growing research area is the use of platelets as a source of therapeutic extracellular vesicles (EV) [10]. EV are membrane-bound lipid vesicles that mediate cell-to-cell communication by delivering bioactive cargo such as proteins, lipids, nucleic acids, and signaling molecules [11]. EV encompass three main subgroups of vesicles that differ in biogenesis, size and cargo: exosomes (40–120 nm), microvesicles (50–1000 nm), and apoptotic bodies (500–2000 nm) [12,13,14]. These EV can be released naturally by all cell types under physiological conditions, but also in response to microenvironmental changes (e.g., cellular stress or disease) [12]. In fact, PC have several advantages as sources of EV, including well-defined and regulated collection and characterization procedures, recognized clinical value, and production in a concentrated form [10].

Platelet-derived extracellular vesicles (pEV) were first reported by Peter Wolf, who suggested their release from platelet granules during storage [15,16]. pEV represent the most abundant EV in the human blood of healthy individuals, accounting for 70–90% of the total EV in the bloodstream [17]. Concerning the therapeutic potential of pEV, recent trends have investigated the beneficial use of pEV over conventional carriers as drug delivery systems [18], mainly for cancer treatment. The scientific rationale behind the targeted interaction of pEV with cancer cells lies in the vast range of surface receptors and glycoproteins, such as platelet P-selectin (CD62p), GPIIb/IIIa, or PECAM-1 (platelet-endothelial cell adhesion molecule-1), present on the surface of pEV [19]. The targeting ability of platelet membrane receptors has received interest as a drug delivery system capable of substituting synthetic nanocarriers, which have several disadvantages in terms of targeting ability, safety, biodistribution, biocompatibility, and tracking [20]. Engineered nanocarriers with encapsulated anti-cancer drugs, functionalized with whole platelet membranes [21,22,23] or with a few platelet membrane proteins [24,25], have shown excellent targeting performance, the ability to escape immune clearance, and inhibition of drug resistance, tumor growth, and metastasis [26]. However, difficulties in the isolation of platelet membranes and the functionalization of synthetic nanocarriers favor the use of pEV as drug carriers instead [26].

In 1963, following screening for anti-cancer activity, paclitaxel (PTX) was discovered, posteriorly approved for clinical trials, and employed in cancer therapies as the first-line therapeutic agent in breast or ovarian cancers [27]. Despite its potential, PTX has important aspects that should not be overlooked when seeking additional chemotherapeutic treatments. PTX has highly poor aqueous solubility, resulting in low bioavailability and non-selective toxicity, leading to several adverse effects in vivo [28]. Therefore, selection of a novel delivery mechanism to package PTX is required to significantly reduce its side effects and improve its therapeutic outcomes. In the current study, we explored for the first time the potential of using pEV as an advanced PTX drug delivery system for targeted treatment of breast cancer. In this context, we separated pEV from PC by exploring different physical-based protocols to identify the most efficient. We then loaded the pEV with PTX and assessed their effects on angiogenesis, cell migration, and invasion.

## 2. Materials and Methods

### 2.1. Platelet Concentrates Collection, Quality Control, and Processing

PC collected from healthy volunteer donors were provided by the Portuguese Institute for Blood and Transplantation (IPST, Lisbon, Portugal), 1–2 days after the clinical expiration date. Blood donations were collected and processed in compliance with Portuguese and European regulatory legislation [29,30]. Blood donations of all ABO and RhD blood groups were screened for infectious disease markers on a regular basis (human immunodeficiency virus, hepatitis C virus, hepatitis B virus, and *Treponema pallidum*). One PC consisted of pooling four buffy coats or platelet-rich plasma blood group-compatible units. The obtained PC were re-suspended in 300–450 mL of a medium containing 30% (*v*/*v*) plasma and 70% (*v*/*v*) platelet additive solution (InterSol), yielding a minimum final platelet content of 2.0 × 10^11^ per PC unit. PC were kept at 22 °C (±2 °C) under continuous agitation. Once the shelf-life was reached (5–7 days after collection), the PC were centrifuged for 10 min at 1000× *g* (5010R centrifuge, Eppendorf) at room temperature, and the supernatants were stored at −20 °C.

### 2.2. Flow Cytometry-Based Platelet Activation

Platelet activation was evaluated by detecting the platelet cell surface marker CD62p (P-selectin) using flow cytometry [31]. Platelets were resuspended in Tyrode’s buffer with albumin and glycose (1.0 × 10^8^ platelet/mL) and incubated with fluorochrome-labeled anti-CD61 and anti-CD62p antibodies. Peridinin chlorophyll protein-conjugated anti-CD61 was used to determine the total number of platelets, and the percentage of activated platelets was determined using phycoerythrin-conjugated anti-CD62p (P-selectin; BD Biosciences, Franklin Lakes, NJ, USA). After 15 min of incubation in the dark at room temperature, flow cytometry was performed using a FACSCalibur^TM^ Cytometer (Becton Dickinson) and Cell Quest software. In tandem, platelets were agonist-stimulated for 15 min with thrombin (0.05 U/mL) resuspended in Tyrode’s buffer containing albumin and glycose, in the presence of Gly-Pro-Arg-Pro (GPRP; BD Biosciences) to abrogate platelet aggregation. Activated platelets were defined as the percentage of CD61-positive events co-expressing CD62p.

### 2.3. pEV Separation from Platelet Concentrates

To remove remaining residual platelets, debris, and pEV aggregates, PC supernatants that had been kept at −20 °C were thawed overnight at 4 °C, centrifuged at 2000× *g* (5010R centrifuge, Eppendorf) for 10 min at room temperature, and then filtered through 0.45 μM filter units (Nalgene^TM^ Rapid-Flow^TM^, ThermoFisher Scientific, Waltham, MA, USA), as previously described, with minor modifications [32,33]. Filtered supernatants were placed in 30 mL conical open-top polypropylene tubes (Beckman Coulter, Brea, CA, USA) and ultracentrifuged at 110,000× *g* for 2 h 45 min at 4 °C (SW 28 rotor, Ultracentrifuge Optima^TM^ LE-80K, Beckman Coulter). The generated pEV pellets were used as the starting material for the pEV isolation protocols. Three methods were used and compared as described below.

#### 2.3.1. pEV Isolation by Discontinuous Iodixanol Density Gradient Ultracentrifugation (DGUC)

A discontinuous 40–5% iodixanol gradient was prepared from OptiPrep™ density gradient medium [60% (*w*/*v*) aqueous iodixanol solution, BioVision], as previously reported [34], with some modifications [33]. A 50% (*w*/*v*) iodixanol working solution was prepared by diluting 60% (*w*/*v*) OptiPrep^TM^ stock solution in sucrose buffer [60 mM tromethamine-hydrochloride acid (Tris-HCl), 6 mM ethylenediamine tetraacetic acid (EDTA), 0.25 M sucrose, pH 7.4]. To prepare iodixanol 5, 10, and 20% (*w*/*v*) gradient solutions, adequate quantities of 50% (w/v) working solution were mixed with a homogenization buffer (10 mM Tris-HCl, 1 mM EDTA, and 0.25 M sucrose, pH 7.4). The pEV pellet was suspended in a 50% (*w*/*v*) working solution to prepare a 40% (*w*/*v*) iodixanol solution. The gradient solutions were layered in open-top polypropylene tubes (Beckman Coulter) to form a discontinuous 40–5% iodixanol gradient consisting of 4 mL layers of 10, 20 and 40% (*w*/*v*), 3.5 mL of 5% (*w*/*v*), and 1 mL of DPBS. The samples were ultracentrifuged at 110,000× *g* for 18 h at 4 °C (SW28 rotor, Ultracentrifuge Optima^TM^ LE-80K, Beckman Coulter). Gradient fractions of 1 mL were carefully collected from top to bottom and pooled according to the subsequent ranges: 1–4, 5–7, 8–9 (pEV containing fractions, interfacing 1.08 g/mL–1.13 g/mL densities), 10–12, 13–16. Pooled gradient fractions were concentrated using Amicon^®^ Ultra-2 mL 10 KDa filter units (Merck Millipore, Burlington, MA, USA) and stored at −80 °C, until use. As a negative control gradient, the same isolation procedure was performed with DPBS (Gibco, ThermoFisher Scientific) (DPBS gradient), instead of pEV pellet.

#### 2.3.2. pEV Isolation by Size Exclusion Chromatography (SEC)

A 70 nm qEV original column packed with a polysaccharide resin (Izon Science) was used according to the manufacturer’s instructions. Briefly, the column was equilibrated with DPBS and 500 μL of the pEV pellet resuspended in DPBS was loaded on top of the column. A total of 17 sample fractions of 500 μL volume each were collected immediately after sample loading. The fractions 1–5, 6–7, 8–9, 10–13, and 14–17 were pooled, further concentrated using Amicon^®^ Ultra-2 mL 10 KDa filter units (Merck Millipore), and stored at −80 °C until use in downstream experiments.

#### 2.3.3. pEV Isolation by DGUC Followed by SEC (DGUC-SEC)

The pEV-enriched fractions obtained by DGUC (Section 2.3.1) were applied to a qEV/70 nm column and processed as described above (Section 2.3.2).

### 2.4. Paclitaxel Loading into pEV by Direct Incubation

Paclitaxel (PTX) loading into pEV was performed as previously described by Kim et al., with minor modifications [35]. In brief, 50 μM (64 μg/mL) of PTX or DMSO vehicle (Dimethyl sulfoxide, CryoSure-DMSO, WAK-Chemie Medical GmbH) were mixed with 5 × 10^10^ pEV in DPBS. Samples were incubated for 1 h at 37 °C with constant agitation at 350 rpm (ThermoMixer^®^ C Eppendorf). Free drug was removed using DGUC (Section 2.3.1). The resulting PTX-pEV samples were used immediately for characterization or functional experiments, or stored at 4 °C until further use.

A calibration curve (Appendix A) for the indirect quantification of PTX loaded into pEV was prepared according to previous publications [36,37]. The PTX stock solution was diluted with 30% (*v*/*v*) methanol in DPBS to achieve concentrations between 2.0–20.0 μg/mL. The calibration values were measured by UV at a wavelength of 230 nm (Shimadzu UV-1603 spectrophotometer) with correction for the blank. Once PTX was dissolved in DMSO, the absorbance values were corrected against the corresponding values in DMSO.

To measure PTX concentration in pEV, approximately 2 × 10^9^ PTX-pEV were treated with 100 μL of 1 × RIPA buffer (Sigma) and placed on a thermomixer (ThermoMixer^®^ C Eppendorf) for 30 min at 4 °C with shaking (650 rpm). Subsequently, the absorbance was measured as previously described [38].

### 2.5. Protein Extraction and Western Blot Analysis

Previously isolated pEV and PC samples were lysed in RIPA buffer (Sigma) containing an EDTA-free protease inhibitor cocktail [Roche, 1:25 (*v*/*v*)]. The total protein content of platelets and pEV was quantified in triplicate using the Pierce^TM^ BCA Protein Assay kit (ThermoFisher Scientific), according to the manufacturer’s instructions.

For Western Blot analysis, platelet lysates and pEV lysates with equal amounts of protein (20 μg) or equal volumes were denatured by boiling for 5 min at 95 °C (ThermoMixer^®^ C Eppendorf) in 4 × lithium dodecyl sulfate sample buffer (Novex^®^ ThermoFisher Scientific) with 10 × NuPAGE^®^ reducing agent (Novex^®^ ThermoFisher Scientific) for antibodies anti-Flotillin-2, anti-Argonaute-2, and anti-APOA1, or without for antibodies anti-CD9, anti-CD41, and anti-CD63. Protein samples were loaded onto SDS polyacrylamide gel electrophoresis (Novex^®^ ThermoFisher Scientific) at 200 V for one hour in MOPS SDS Running buffer (Novex^®^ ThermoFisher Scientific). Proteins were transferred to polyvinylidene fluoride membranes (iBlot 2 Transfer Ministack, Invitrogen, Waltham, MA, USA) via dry electroblotting using the iBlot™ 2 Dry Blotting System (Invitrogen). Membranes were blocked with 5% (*w*/*v*) of non-fat dry milk (PanReac AppliChem, Chicago, IL, USA) in Tris-buffered saline + 0.1% (*v*/*v*) Tween 20 (Merck) (TBS-T) for 1 h at room temperature. After incubation with primary antibodies (anti-Flotillin-2: BD Biosciences, ab134131, 1:1000; anti-Argonaute-2: Abcam, ab32381, 1:1000; anti-APOA1: Abcam, ab20453, 1:1000; anti-CD9: Invitrogen, 10626D, 1:1000; anti-CD41: Abcam, ab134131, 1:1000; anti-CD63: Abcam, ab59479, 1:1000), the membranes were thoroughly rinsed with TBS-T and incubated with the appropriate secondary antibody (anti-rabbit-HRP: System Biosciences, EXOAB-KIT-1, 1:10,000; anti-mouse-HRP, Amersham, 1:5000) for 1 h at room temperature. Chemiluminescent signals were detected using the WesternBright^®^ ECL (enhanced chemiluminescence detection, Advansta) and pictures were taken using the ChemiDoc XRS+ System (Bio-Rad Laboratories, Hercules, CA, USA).

### 2.6. Nanoparticle Tracking Analysis (NTA)

The size distribution profile and concentration of pEV in the enriched fractions were tracked by NTA using a NanoSight NS300 system (Malvern Instruments, Morvern, UK) equipped with a blue laser light source (405 nm, 70 mW) and a high sensitivity sCMOS camera. The samples were diluted in 0.2 μm filtered DPBS to obtain a concentration within the optimal range (3 × 10^8^ − 1 × 10^9^ particles/mL). Particle measurements were generated by the analysis of 3 videos of 60 s each acquired at 25 °C, using the NanoSight NTA 3.3 software (Malvern Panalytic Inc., Westborough, MA, USA) with an optimized detection threshold of 5 and a screen gain of 10.0.

### 2.7. Transmission Electron Microscopy (TEM) of pEV

pEV were mixed with 4% (*w*/*v*) formaldehyde in 0.1 M phosphate buffer saline (pH 7.4), incubated for 5 min at room temperature, and adsorbed onto 100 mesh formvar-carbon coated copper grids. The grids were washed with distilled water and negative-stained with 2% (*w*/*v*) uranyl acetate for 5 min at room temperature. Grids were imaged with a Tecnai^TM^ G^2^ Spirit BioTWIN (FEI Company, Hillsboro, OR) transmission electron microscope operating at 120 kV. Images were recorded using a digital charge-coupled device camera (Olympus-SIS Veleta, Münster, Germany).

### 2.8. Transmission Electron Microscopy (TEM) of Platelets

Platelet pellets were fixed by adding 2.5% glutaraldehyde (*w*/*v*) and 1.25% (*w*/*v*) formaldehyde in 0.1 M phosphate buffer for 1 h at 4 °C. After carefully removing the fixative, platelets were embedded in 2% (*w*/*v*) agarose and allowed to clot at 4 °C. The clotted specimens were cut and cubes of approximately 1-mm were post-fixed in 1% (*v*/*v*) osmium tetroxide and 1.5% potassium ferrocyanide (*w*/*v*) in 0.1 M phosphate buffer for 1 h at 4 °C. After washing, specimens were stained with 1% (*v*/*v*) uranyl acetate for 1 h at room temperature in the dark. Following staining, the specimen was passed through ascending gradual series of ethanol concentrations, starting at 30% for 10 min, followed by 50% for 10 min, 70% for 10 min, 90% for 10 min, and 100% for 10 min, three washes per cycle, to complete specimen dehydration. Subsequently, infiltration was initiated by submerging the specimen in a series of EPON-resin (EMbed 812 Kit, Electron Microscopy Sciences, Hatfield, PA, USA) mixtures at different concentrations to remove ethanol. After infiltration was complete, the specimen was embedded with 100% resin overnight in an oven at 60 °C for polymerization. Ultrasections of approximately 0.8-μm-thickness were cut and examined in a Tecnai^TM^ G^2^ Spirit BioTWIN (FEI Company, Hillsboro, OR, USA) at 120 kV. Digital images were captured using an Olympus-SIS Veleta digital charge coupled camera.

### 2.9. Human Umbilical Vein Endothelial Cells (HUVEC) Culture

Human umbilical vein endothelial cell (HUVEC) line (CC-2517A, Lonza, Basel, Switzerland) was cultured in endothelial cell growth medium (EBM^TM^-2 Basal Medium, Lonza) supplemented with EGM^TM^-2 Endothelial SingleQuots^TM^ Kit. Cells were maintained at 37 °C in a saturated humidified atmosphere containing 5% (*v*/*v*) CO_2_ and 95% (*v*/*v*) air. The culture medium was exchanged every 2–3 days. Cells were harvested with TrypLE^TM^ Select (Gibco, ThermoFisher Scientific) and subcultured upon reaching 70–85% confluency. Cells were used for experiments up to passage 5.

### 2.10. Breast Cancer Cells (MDA-MB-231 and BT474) Culture

Luminal B BT474 (ATCC HTB-20) (ER^+^, PR^+^, HER2^+^) and triple-negative MDA-MB-231 (ATCC HTB-26) (ER^−^, PR^−^, HER2^−^) human breast cancer cell lines were cultured in phenol red-free RPMI-1640 culture medium (Gibco, ThermoFisher Scientific) supplemented with 10% (*v*/*v*) fetal bovine serum (Gibco, ThermoFisher Scientific). These cells were maintained at 37 °C in a humidified atmosphere with 5% (*v*/*v*) CO_2_ and the medium was replaced every 2–3 days. Once cells reached 80–90% confluency, they were harvested with TrypLE^TM^ Select (Gibco, ThermoFisher Scientific) and subcultured according to functional assays.

### 2.11. Cellular Uptake of PKH26-Labelled pEV

Purified pEV were stained with PKH26 (PKH26-pEV) lipophilic red fluorescent membrane dye (Red Fluorescent Cell Linker for General Cell Membrane Labeling Mini kit, Sigma-Aldrich), as previously described, with minor modifications [39]. Briefly, prior to staining, 50 μL of diluent C was added to 50 μL of the concentrated pEV stock and mixed gently by pipetting. After that, 1.5 μL of PKH26 was diluted in 100 μL of diluent C, and pEV were added to the diluted PKH26. The mixture was incubated for 5 min at room temperature protected from light, and the labeling reaction was blocked by adding 100 μL of 0.1% (*v*/*v*) ultracentrifuged bovine serum albumin (BSA, Gibco, ThermoFisher Scientific). To remove unincorporated stains, the mixture was subjected to DGUC as described in Section 2.3.1. The protocol was repeated using DPBS as the negative control (DPBS-PKH26). The resulting PKH26-labeled samples were immediately added to the cell cultures or stored at 4 °C until further use.

To investigate the kinetics of pEV internalization, PKH26-pEV were incubated with HUVEC and breast cancer cell lines. HUVEC, MDA-MB-231, and BT474 cells were seeded in 24-well plates containing gelatin pre-coated glass coverslips at densities of 1.3 × 10^4^, 2.0 × 10^4^, and 2.5 × 10^4^ cells/cm^2^, respectively. After 24 h of culture, fresh fully supplemented media containing 6000 PKH26-pEV/cell or an equal volume of DPBS-PKH26 (negative control) were added to cells. Cells were fixed in 4% (*w*/*v*) paraformaldehyde for 15 min at room temperature and blocked with 0.2% (*w*/*v*) fish skin gelatin. HUVEC were then stained for CD31 (anti-CD31: Agilent Technologies, M0823, 1:50), MDA-MB-231 for EGFR (anti-epidermal growth factor receptor, anti-EGFR: Cell Signaling Technology, D38B1, 1:100), and BT474 for HER2 (anti-human epidermal growth factor receptor 2, anti-HER2: Abcam, ab134182, 1:100) overnight and nuclei were counterstained with DAPI (4′,6-diamidino-2-phenylindole, dihydrochloride, ThermoFisher Scientific). Coverslips were mounted using ProLongTM Gold Antifade reagent (Molecular Probes, Invitrogen). The samples were visualized using an inverted fluorescence microscope (DMI6000, Leica, Wetzlar, Germany).

For flow cytometry analysis, HUVEC, MDA-MB-231, and BT474 cells were seeded at a density of 5.2 × 10^4^ cell/cm^2^ (6-well plates). On the following day, cells were incubated with PKH26-pEV (6000 pEV/cell) or DPBS-PKH26 in equivalent volume for 24 or 48 h at 37 °C in an atmosphere containing 5% (*v*/*v*) CO_2_. Cells were dissociated with Tryple^TM^ Select at room temperature for 5 min, centrifuged at 300× *g* for 5 min, washed twice with DPBS, and resuspended in 2% (*v*/*v*) FBS in DPBS. Samples were analyzed using a BD FACSCelesta^TM^ flow cytometer (BD Biosciences), collecting a minimum of 20.000 events. Data was acquired using BD FACS DIVA software and analyzed using FlowJo software (TreeStar, Woodburn, OR, USA).

### 2.12. Tube Formation Assay

The ability of PTX-pEV to affect the formation of tubule-like structures in HUVEC was determined using a tube formation assay. HUVEC were seeded in 6-well plates at a density of 2.0 × 10^3^ cell/cm^2^. One day after, cells were serum-starved with 0.1% (*v*/*v*) FBS EGM^TM^-2 basal medium with pEV, PTX-pEV, and vehicle (DMSO-pEV) for 24 h. HUVEC were harvested and seeded at a density of 3.8 × 10^4^ cell/cm^2^ in 96 well-plates previously coated with 40 μL/well of ice-cold growth factor-reduced Matrigel solution (Corning^®^ Matrigel^®^ Growth Factor Reduced Basement Membrane Matrix, Phenol red-free, 356,231). After pre-incubation at 37 °C for at least 30 min, cells were treated with the positive control (fully supplemented EBM^TM^-2), and the subsequent conditions were properly diluted in starvation media: pEV, 8–9 fractions of DPBS gradient (negative control), PTX-pEV, DMSO-pEV (vehicle control), PTX as free drug and free DMSO. Plates were placed in IncuCyte ZOOM^®^ (Essen Bioscience, Ann Arbor, MI, USA) for up to 8 h, and images of tubular structures were taken every two hours (five images per well were acquired). The total segment length and number of nodes were determined using the Angiogenesis Analyzer plugin of ImageJ software [40].

### 2.13. Scratch Wound Assay

A scratch wound assay with HUVEC, MDA-MB-231, and BT474 cultures was used to determine whether PTX-pEV could negatively impact cell migration in vitro. HUVEC were seeded at a density of 9.4 × 10^4^, MDA-MB-231 at 1.3 × 10^5^, and BT474 at 1.6 × 10^5^ cell/cm^2^ in 96-well ImageLock^TM^ microplates (Essen Bioscience) and incubated for 24 h in complete medium until a 95–100% confluent monolayer was reached. HUVEC monolayers were serum-starved in EBM^TM^-2 basal medium (Lonza) with 0.1% (*v*/*v*) exosome-depleted FBS (Gibco, ThermoFisher Scientific), and breast cancer cells in RPMI (Gibco, ThermoFisher Scientific) with 0.5% (*v*/*v*) exosome-depleted FBS (Gibco, ThermoFisher Scientific) for 24 h. Simultaneously, the cells were pre-treated with pEV-tested conditions (pEV, PTX-pEV, and DMSO-pEV). After 24 h of pre-treatment under serum-starvation conditions, a straight scratch was generated in the middle of the wells using a 96-pin mechanical wound-making device (WoundMaker^TM^ Essen Bioscience) to allow cell culture to migrate over mimicking healing. The scratch-wounded cells were washed twice with DPBS to remove cell fragments or detached cells before incubating in fully supplemented EBM^TM^-2 and RPMI (positive control) or in fresh starvation media under the tested conditions (8–9 fractions of DPBS gradient (negative control), pEV, PTX-pEV, DMSO-pEV, PTX as free drug, and free DMSO). Plates were placed in IncuCyte ZOOM^®^ (Essen Bioscience) and cell migration was automatically monitored every two hours until the wounds closed. The scratch area was determined with the aid of a wound-healing size plugin [41] using ImageJ software (NIH, Bethesda, MD, USA). The percentage of wound closure was estimated using Equation (1):(1)$$Wound\;Closure\left(\%\right)=\left(\frac{A_{0}-A_{t}}{A_{t}}\right)\times 100\%$$
where *A*_0_ is the initial wound area, and *A_t_* is the wound area after time *t* of the initial scratch.

### 2.14. Cell Invasion Assay

To detect the impact of PTX-pEV on the invasive ability of MDA-MB-231 cells, a matrigel invasion assay was performed. ImageLock^TM^ 96-well microplates (Essen Bioscience) were pre-coated with 50 µL of 100 µg/mL growth factor-reduced matrigel (Corning^®^ Matrigel^®^ Growth Factor Reduced Basement Membrane Matrix, Phenol red-free, 356,231) per well. After overnight incubation at 37 °C, the remaining matrigel solution was discarded from each well, and cells were seeded (1.3 × 10^5^ cells/cm^2^) in 10% (*v*/*v*) FBS RPMI for 24 h. Cells were then serum-starved overnight with 0.5% (*v*/*v*) exosome-depleted FBS (Gibco, ThermoFisher Scientific) and pre-treated overnight with RPMI with pEV, PTX-pEV, or vehicle (DMSO-pEV). Wounds were created with 96-well WoundMaker^TM^ (Essen Bioscience), and 500 µL of 8 mg/mL matrigel diluted in starvation medium under the following treatment conditions were added to each well: pEV, PTX-pEV, DMSO-pEV, PTX as free drug, and free DMSO. Once the plate was equilibrated on a pre-chilled CoolBox 96F (Essen), 100 µL of additional starvation medium under treatment conditions was added to each well. Plates were placed in an IncuCyte ZOOM^®^ (Essen Bioscience) incubator, and the scratch area was monitored for 48 h, with repeated scanning every two hours. Relative wound density (%) was calculated using IncuCyte software (IncuCyte S3 Software update 2018B) according to Equation (2):(2)$$Relative\;Wound\;Density\left(\%\right)=\left(\frac{w_{t}-w_{0}}{c_{t}-w_{0}}\right)\times 100\%$$
where *w*_0_ is the density of initial wound region, *w_t_* is the density of wound region after period *t* of the initial scratch, and *c_t_* is the density of cell region at time *t*.

### 2.15. Statistical Analyses

All data are presented as mean ± standard deviation (*n* = 3 biological replicates, except where stated. *n* represents the number of independent experiments performed-indicated in figure legends). Statistical significance was determined by Student’s *T* test and one-way analysis of variance (ANOVA) with Tukey’s multiple comparison test using GraphPad Prism 8.0 software. * *p* < 0.05, ** *p* < 0.01, *** *p* < 0.001, **** *p* < 0.0001 were considered significant.

## 3. Results

### 3.1. Characterization of pEV Separated by an Iodixanol Density Gradient of Platelet Concentrates

To characterize expired PC as the starting material to isolate pEV (Figure 1A), platelet activation and reactivity to physiological agonists (thrombin) were assessed. The basal expression level of P-selectin (CD62p) in expired platelets revealed that pEV were isolated from highly stimulated platelets (Figure 1B), which is likely linked to the storage period of PC, as previously reported [15]. Therefore, to evaluate platelet functional efficiency, the expression of CD62p on the surface of platelets was examined after addition of thrombin. Our results showed a statistically significant increase in the measured activation level of agonist-treated blood platelets (Figure 1B), indicating that platelets, even when initially highly activated, responded to agonists, demonstrating their functionality. The quality of the expired platelets was also examined based on their ultrastructure and degree of maturity. Transmission electron microscopy (TEM) analysis confirmed that platelets displayed typical size heterogeneity and a discoid or spherical shape (Figure 1C). Platelets with elongated cell membrane projections and with a decrease in platelet size compared to resting platelets were also observed, which is typical of activated platelets [15,42].

Since blood is the most complex body fluid and PC are blood-derived products, the choice of isolation method is an essential factor for achieving purified and functional pEV [43,44]. Due to a lack of consensus among the scientific community [45], we compared three pEV isolation techniques that explored distinct physiochemical properties of EV, namely, iodixanol density gradient ultracentrifugation (DGUC), size-exclusion chromatography (SEC), and a combination of both approaches, i.e., DGUC followed by SEC (DGUC-SEC). The higher pEV yield and residual co-isolation of non-EV impurities led to the selection of DGUC as the optimal isolation method (Appendix A). In fact, DGUC allowed the isolation of approximately 2 and 12 times more particles than SEC and DGUC-SEC, respectively. In the DGUC protocol (Figure 1A), pooled 8–9 fractions (interfacing 1.08 g/mL–1.13 g/mL densities) were selected as pEV-rich fractions because they provided the best balance of yield and purity and were used in further characterizations.

A Western blot assay was performed to confirm the enrichment of specific EV markers in pEV samples isolated by DGUC method. In pooled 8–9 fractions (interfacing 1.08 g/mL–1.13 g/mL densities), enriched expression levels of the common EV protein markers CD63 (Figure 1D; Appendix A), CD9 (Figure 1D; Appendix A), and flotillin-2 (FLOT2, Figure 1D; Appendix A) were observed. The presence of platelet-specific surface marker CD41 (Figure 1D; Appendix A) in the pEV-enriched fractions confirmed the cellular origin of the isolated EV. The non-EV markers apolipoprotein A1 (ApoA1) and argonaute-2 (Ago2) were used to detect the presence of high-density lipoproteins (HDL) and RNA-binding protein contaminants, respectively, and were absent in 8–9 fractions (Figure 1D and Appendix A). TEM analysis of pooled 8–9 fractions confirmed the presence of vesicles with the cup-shaped morphology that is typically observed in electron microscopy images of EV (Figure 1E). Nanoparticle Tracking Analysis (NTA) revealed a bimodal size distribution profile ranging from 100 to 300 nm (Figure 1F). In particular, the main peak was at 148 nm and the lower peak was at 202 nm, indicating that two main pEV populations were isolated.

### 3.2. Cellular Uptake of pEV by Endothelial and Breast Cancer Cells

We further investigated the bioactivity of pEV upon uptake by endothelial cells (HUVEC) and breast cancer cells (MDA-MB-231 and BT474) at various time points (24 and 48 h). It is important to note that a dose of 6000 particles per cell was applied in the functional assays, which was identified through a dose-response curve as the optimal dose to elicit an effect in recipient cells (Appendix A). To label and track pEV, a lipophilic membrane dye (PKH26) was used. Previous research has found that simple washing by sedimentation was ineffective in removing dye clumps, whereas sucrose density-gradient ultracentrifugation showed its effectiveness [46]. Therefore, to avoid non-specific fluorescence signals, excess unbound or aggregated dyes were removed using the DGUC procedure. To validate that the observed results were from PKH26 labeled pEV (PKH26-pEV), a DPBS-PKH26 negative control was performed in parallel. The formation of two dye bands on the density gradient in PKH26-pEV samples was observed: one at the level of 8–9 fractions corresponding to PKH26-pEV, and a second band on denser fractions, representing the unbound dye (Appendix A). Only the band on denser fractions of the DPBS-PKH26 negative control was found, indicating the ability of DGUC to separate PKH26-pEV from unbound dye and dye aggregates (Appendix A). After incubating the cells with PKH26-pEV for 24 h, punctate fluorescent signals were imaged in the recipient cells, which were mainly located in the cell periphery and perinuclear regions (Figure 2A).

Notably, the lack of fluorescence in cells incubated with DPBS-PKH26 (Appendix A) confirmed the effective removal of unbound PKH26 or PKH26 aggregates. Flow cytometry analysis revealed that PKH26-pEV were internalized by recipient cells after 24 h and 48 h of incubation in a time-dependent manner, with uptake increasing from 79.0%, 98.2%, and 68.9% at 24 h to 99.4%, 99.3%, and 87.9% at 48 h in HUVEC, MDA-MB-231, and BT474 cells, respectively (Figure 2B).

### 3.3. Paclitaxel Loading and Entrapment Efficiency

The incorporation of PTX into pEV (PTX-pEV) was performed through direct incubation by exploiting the high hydrophobicity of PTX [47]. In brief, 5 × 10^10^ pEV were incubated with 64 μg/mL PTX for 1 h at 37 °C (Figure 3A). Likewise, the same loading approach was employed for pEV incubated with DMSO to confirm that the results obtained were not mediated by the vehicle solution. After removal of the unbound drug by DGUC (Figure 3A), cargo loading was measured by UV-VIS spectrophotometry, as previously described [48].

An entrapment efficiency of 5.92 ± 0.01% of EV-bound PTX was observed, with an average PTX concentration of 3.79 μg/mL ± 0.04 in pEV preparations. Considering that drug loading could have a significant impact on vesicle size and morphology, the integrity of the vesicles was evaluated following PTX incorporation [49]. TEM micrographs showed no morphological changes in samples after PTX or DMSO loading (DMSO-pEV) (Figure 3B). Additionally, no significant differences in the mode and mean particle sizes were observed before and after drug loading (Figure 3C,D).

To confirm PTX entrapment in pEV, cellular uptake of pEV loaded with tagged PTX (PTX488-pEV) was performed in HUVEC. Microscopic images of PTX448-pEV and PTX448-pEV labeled with PKH26 (PTX488-PKH26-pEV) revealed a significant degree of drug colocalization with pEV, demonstrating the successful entrapment of PTX within pEV (Appendix A).

### 3.4. Antiangiogenic Effects of Paclitaxel-Loaded pEV

Considering that angiogenesis promotes tumor growth and progression, and PTX has been described as an angiogenic inhibitor [50,51], the potential of pEV as paclitaxel-delivery vehicles (PTX-pEV) to suppress cancer progression action was examined. An in vitro tube formation assay was conducted to study the effects of PTX-pEV on the tube-like network formation capacity of HUVEC. To ensure that observations from PTX-pEV were caused by the presence of PTX, two control conditions were used in this study: DMSO-pEV and pEV. As seen by the uneven and broken tubes (Figure 4A), treatment with PTX-pEV inhibited the endogenous formation of tridimensional vessels in vitro, as measured by a reduction in the overall segment length by 33 ± 5% (*p* value 0.0093) (Figure 4B) and the number of nodes by 56 ± 16% (*p* value 0.1311) (Figure 4C) compared to the vehicle control (DMSO-pEV). Furthermore, PTX-pEV (estimated concentration of 0.4 μM) exhibited an impact on all angiogenic parameters, statistically similar to free drug treatments (0.5 μM PTX and 1 μM PTX) (Figure 4B,C).

### 3.5. Antimigratory Effects of Paclitaxel-Loaded pEV

Next, we investigated the effects of PTX-pEV treatment on the migratory ability of endothelial and breast cancer cells using a wound healing assay. As shown in Figure 5, clear inhibition of the wound closure ability of HUVEC (Figure 5A,B), MDA-MD-231 (Figure 5C,D), and BT474 (Figure 5E,F) was observed after 16, 24, and 48 h of PTX-pEV treatment, respectively. For BT474, 48 h of incubation with PTX-pEV was performed, whereas 24 h were established for MDA-MB-231 cells (Figure 5C,D). The rationale for choosing a shorter incubation time correlated with the metastatic potential of MDA-MB-231 because of its increased migratory ability compared with other breast cancer cell lines [52]. At the end of the follow-up period, results showed that PTX-pEV significantly inhibited HUVEC migration by ~66% (*p* value 0.0381), MDA-MB-231 by ~44% (*p* value 0.0756), and BT474 migration by ~34% (*p* value 0.4138), compared to the DMSO-pEV control (Figure 5B,D,F). Additionally, the effect of PTX-pEV did not show statistically significant differences to 0.5 μM free PTX condition in all cell lines (HUVEC: 22.9 ± 3.1% for 0.5 μM PTX and 20.8 ± 8.9% for pEV, Figure 5B; MDA-MB-231: 26 ± 3% for 0.5 μM PTX and 44 ± 11% for pEV, Figure 5D and BT474: 38 ± 13% for 0.5 μM PTX and 37 ± 10% for pEV, Figure 5F).

### 3.6. Anti-Invasive Effects of Paclitaxel-Loaded pEV

MDA-MB-231 is a triple-negative breast cancer cell line that lacks the expression of estrogen receptor (ER^−^), progesterone receptor (PR^−^), and human epidermal growth factor receptor 2 (HER2^−^). This subset of breast cancer cells exhibits aggressive clinical behavior and an invasive phenotype in vitro [53]. In this way, we were interested in investigating the role of PTX-pEV in MDA-MB-231 invasion capacity, since invasiveness is an important feature of tumor progression.

A matrigel invasion assay was performed to assess the anti-invasive effect of PTX-pEV on MDA-MB-231 cells. Cell invasion was dramatically reduced after 48 h of treatment with PTX-pEV (Figure 6A), with only 22 ± 7% area of invasion compared to 51 ± 3% of the DMSO-pEV control (*p* value < 0.0001) (Figure 6B). As expected, treatment with free PTX reduced the ability of MDA-MB-231 cells to invade the monolayer scratch compared to DMSO control (7 ± 1% for 0.5 μM PTX and 67 ± 3% for DMSO, Figure 6B).

## 4. Discussion

Expired platelet concentrates (PC) have emerged as attractive human blood materials with therapeutic applications, as their manufacture by licensed blood establishments ensures compliance with quality requirements [4]. However, the ever-increasing demand for blood platelets, perennial scarcity, and high production costs accentuate the need to develop new products and technologies to reduce PC waste [54]. Human platelets and pEV express membrane receptors that interact with tumor cells and thus have promising applications as targeted anti-cancer delivery systems [10,55]. However, most published studies have focused on the development of synthetic nanocarriers functionalized with platelet membrane proteins [21,24], which are impractical for clinical applications because of their inherent limitations [26]. In addition, considering that breast cancer is one of the most prevalent cancers worldwide, with current treatment strategies restricted by the mechanism of acquired resistance described for existing drugs [56], we studied the potential of pEV as a delivery vehicle for PTX, a first-line therapeutic agent in breast cancer [27]. Through a variety of functional assays with cells involved in the breast tumor microenvironment, we provide evidence that pEV isolated from expired PC constitute a potential delivery vehicle for PTX, with similar therapeutic benefits to the free drug but with the advantage of being a natural carrier.

PC are one of the most expensive blood products, not only because of the optimal conditions required for platelet collection and processing, but also because of the financial costs required to maintain PC quality during storage time. The PC used in this study were maintained at an appropriate storage temperature, pH, bag plastic containers, and agitation profile to ensure the maintenance of its therapeutic attributes. During their storage period, platelets can be easily activated, resulting in structural changes and the release of pEV from multivesicular membranous sacs of platelet pseudopods [17]. Indeed, TEM analysis of platelet samples revealed a heterogeneous population of platelets with activated spherical cells and pseudopods, which correlated with the expression of the activation marker CD62p measured by flow cytometry. These data are in agreement with a previous report, which demonstrated a spontaneous increase in CD62p expression during PC storage period, with four times more platelets activated on day 7 (past shelf-life) relative to day 1 [57]. Thus, expired PC, composed of highly activated platelets, represent an enriched source of pEV. However, the final EV preparation is not entirely composed of pEV, since PC contained 30% plasma. Nevertheless, to induce even more platelet activation and possibly pEV and growth factors release, pulsed electric field technology can be implemented [6]. For personalized medicine, pEV should ideally be prepared from autologous PC sources, as they minimize the risk of immune rejection by the host; even so, only patients undergoing surgery may have autologous sourcing available [58].

Because blood is one of the most difficult body fluids to isolate EV with high purity owing to the high content of lipid particles (e.g., chylomicrons and lipoproteins) in human plasma and serum, PC represent a challenging starting material [44]. As a result, to isolate pEV, different isolation and purification methods were compared (DGUC, SEC, and DGUC-SEC). The pEV-pooled rich fractions isolated by DGUC were characterized in accordance with the MISEV guidelines [59], which revealed an enrichment of EV markers CD63, CD9, and FLOT2 and the absence of non-EV contaminants, apolipoproteins, and RNA-binding proteins. pEV presented the cup-shaped morphology characteristic of TEM images of vesicles. To examine the natural structure of EV without artifacts created by sample processing, cryo-EM should be employed instead [60]. By observing the representative size distribution profile, two pEV populations were isolated, which could be justified by other studies evidencing the existence of two major populations of pEV, small and large [32,61]. The presence of these two EV subpopulations has also been described for EV from different biological fluids, including conditioned culture media [62]. The combination of two methods (DGUC-SEC), which separate EV based on two different physical properties (size and density), increased pEV purity. Considering this, particles overlapping in diameter with EV but with different densities (e.g., chylomicrons; ρ < 0.930 g/mL and 75–1200 nm) or overlapping in density with distinct sizes (e.g., HDL; ρ = 1.063–1.210 g/mL and 8–16 nm) were expected to be removed [44,63]. Even so, DGUC-SEC revealed a clear decrease in pEV yield, which correlates with previous results [43,64,65], indicating that more labor-consuming protocols (i.e., with more processing steps) negatively impact EV yield. With regard to the SEC protocol, a lower pEV concentration was attained compared with the DGUC protocol. These differences may be associated with pEV losses due to non-specific binding to the SEC column resin. In conclusion, DGUC was selected as the most efficient protocol, because it allowed for higher pEV yields and purity.

According to the literature, compared to synthetic carriers, EV appear to be internalized with higher efficiency by recipient cells, advantage that can be attributed to the endogenous intracellular trafficking mechanisms involved in their cellular uptake [66]. Furthermore, it has been shown that EV from the same cellular source can express different intercellular communications and traffic to different types of recipient cells [25,66]. To further address possible variations in the therapeutic potential of PTX-pEV among recipient cells, it is important to elucidate the kinetics and specificity of pEV uptake into different cells. To confirm the intracellular location of PKH26-pEV after uptake by HUVEC, MDA-MB-231, and BT474 cell lines, the recipient cells were stained for relevant markers. CD31 is a sensitive and specific marker for endothelial cells [67], EGFR is highly expressed in triple negative breast cancer cells (e.g., MDA-MB-231) [68], and HER2 is highly expressed in BT474 cells [69]. We did observe that PKH26-pEV were mainly located in the cell periphery and perinuclear regions, which could be an indicator of endocytic internalization [70]. Consistent with this finding, a previous study by Svensson et al. strongly suggested that EV uptake occurs mainly via endocytosis [71]. Even so, more research should be conducted to study the intracellular trafficking and internalization pathway of pEV by live-cell microscopy and EV uptake inhibitors. Furthermore, using a fluorescent marker for labelling acidic organelles (e.g., endosomes and lysosomes) could be useful for determining whether labeled EV were colocalized with the cell endosomal or lysosomal compartments [72].

One major concern in the application of EV as drug delivery systems is their limited drug loading efficiency [20]. Different drug loading approaches have been described, with the possibility of performing the loading protocol directly on the EV after isolation (e.g., direct incubation, electroporation, or sonication) or during EV biogenesis. In this study, PTX was incorporated via direct incubation, benefiting from the small size and high hydrophobicity of PTX (with very poor solubility, ≤0.4 M), allowing for the spontaneous incorporation of this drug into the hydrophobic inner region of EV lipid bilayers [28]. PTX is one of the drugs that have been efficiently loaded into EV by direct incubation, which is a less time-consuming and simpler loading method [73,74]. Because direct incubation is a mild method, as opposed to transfection, electroporation, ultrasound, extrusion, or freeze-thaw cycles, this loading strategy did not damage the integrity of the vesicles [49]. To complement this, the presence of specific platelet membrane translocation proteins (e.g., CD41 and CD47) could have been evaluated to confirm that pEV retained their platelet-specific proteins after direct incubation with PTX [22]. Consistent with previous reports, we observed an average loading entrapment of 5.9% with no pEV morphological changes [35,73]. Another challenging and critical aspect of translating EV-based therapy is the lack of knowledge regarding EV preservation and storage. The storage of EV becomes even more relevant when EV are used as drug delivery vehicles because EV loading methods may disrupt EV structures and consequently their stability. Despite the scarcity of data on the effect of storage conditions on drug-loaded EV, it has been reported that EV content appears to be more stable at 4 °C when compared to repeated cycles of freezing and thawing [75].

Breast cancer is a diverse and heterogeneous disease. To predict a better therapeutic response to PTX-pEV, two breast cancer cell lines (MDA-MB-231 and BT474) were specifically chosen for this study based on their distinct subtypes [76]. BT474 cells are of luminal origin and have a lower propensity for migration, being less aggressive and invasive, whereas the triple negative MDA-MB-231 is a cell line of basal origin and has worse prognosis and a higher metastatic potential [77]. The mechanism of action of PTX involves the stabilization and dysfunction of cytoskeleton microtubules, ultimately resulting in the loss of several vital cellular functions, such as maintenance of cell shape, cell division, mitosis, and cell motility [27]. Angiogenesis is a major prognostic contributor for cancer progression. Angiogenesis is the mechanism by which new vessels develop from endothelial cells in preexisting capillaries [78]. As the development of new capillaries provides a steady supply of oxygen and nutrients, both critical for tumor growth, we explored the role of PTX-pEV in hampering angiogenic processes [78]. In the current study, we observed the potential of PTX-pEV as an anti-angiogenic treatment, since it inhibits HUVEC migration and reduces the vascular area. In addition, equivalent treatment of tumor cells reduced their migratory and invasive capacities to the same extent as treatment with the free drug. Until recently, traditional platelet drug delivery systems relied on in vitro studies that demonstrated the role of platelet receptors in fostering targeted interactions with breast cancer cells through the development of platelet-inspired synthetic nanoparticle systems [25]. Although pEV as a drug delivery system have been shown to have a similar anti-cancer effect to free PTX, for clinical purposes, several advantages of EV should be considered. EV-based drug delivery has shown inspiring potential in the biomedical field as the next generation of nanomaterials for advanced drug delivery, owing to their prospective benefits regarding synthetic nanomaterials and free drug applications. These include low immunogenicity, intrinsic targeting abilities, and enhanced stability in circulation [20], particularly because of their endogenous origin. Another critical component of employing pEV as drug carriers is that pEV-mediated cancer cell targeting may help mitigate the deleterious effects of free drugs, such as cytotoxicity, lack of biocompatibility, and targeting abilities [20,79]. Extensive studies have revealed that EV carrying anti-tumor drugs are pivotal for the selective delivery of drug to tumors, increasing treatment effectiveness, and overcoming drug resistance. Kim et al. showed that EV carrying anti-tumor drugs increase treatment effectiveness and enhanced cytotoxicity in multidrug-resistant cancer cells treated with PTX-loaded exosomes [35]. Furthermore, Quiao et al. reported that doxil (a chemotherapy drug)-loaded exosomes enhanced therapeutic retention in tumors compared to free doxil [80]. In addition, synthetic silica particles functionalized with membrane-derived vesicles from activated platelets and incorporated with tumor-killing cytokines decreased lung metastasis in a mouse breast cancer metastasis model [21]. Pan et al. showed that liposomes loaded with doxorubicin and functionalized with P-selectin and GPIIb-IIa–like receptors, enhanced drug binding and delivery to MDA-MB-231 cells, suggesting a promising approach for metastasis-targeted drug delivery [25]. In addition to the inherent benefits of pEV, we consider that some unknowns need to be addressed to develop PTX-pEV into recognized therapeutic entities. This includes further studies to examine how the intrinsic physiological characteristics of pEV (e.g., growth factors, miRNA, cytokines) may complicate the predictions of its pharmacodynamics [10].

Although in vitro assays are excellent methods for extrapolating to in vivo situations and studying live cell behavior, particularly those applied in this study, a significant weakness of it is that they fail to accurately replicate the physiological and pathological conditions of cells in an organism [81]. Therefore, to further understand the applicability of multiple aspects of PTX-pEV, from dosing and circulation time to administration route, research employing more advanced three-dimensional cell models (e.g., culture in spheroids using different cell components of the tumor microenvironment [82]) or in vivo models must be conducted as a follow-up study to disclose the interactions and biodistribution of PTX-pEV in more complex biological contexts [81]. Notwithstanding, PTX-pEV dose, toxicity, and biodistribution pattern upon in vivo administration in breast cancer animal models should be examined to determine if pEV are preferentially taken up by cancer cells and if there is a low accumulation in the liver, lung, spleen, and gastrointestinal tract [83].

## 5. Conclusions

Our findings show for the first time that pEV isolated from expired PC could be employed as a tailored targeted drug delivery system for the treatment of breast cancer. An integrated platform for the development of a new therapeutic solution for expired PC was established, beginning with the identification of the most efficient protocol for pEV isolation, culminating in assessing the in vitro effects of pEV loaded with PTX. We found that the DGUC protocol provided pure and functional pEV, that can be used as PTX carriers. We demonstrated that PTX-pEV exhibited in vitro anti-angiogenic, anti-migratory, and anti-invasive activities in different cells potentially present in the breast cancer microenvironment. Despite the need for future studies to better address the mechanisms and pharmacokinetics behind the effects of PTX-pEV, our results are a promising step towards exploring a wasted source of EV in the design of therapeutic platforms for cancer treatment.

## Figures and Tables

**Figure 1 pharmaceutics-15-00953-f001:**
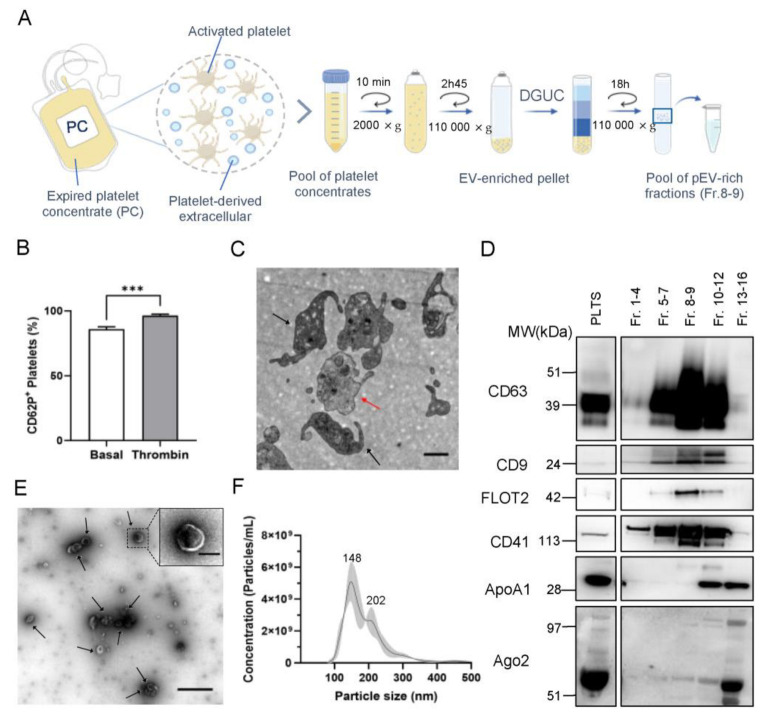
Characterization of platelets and isolated platelet-derived extracellular vesicles (pEV). (**A**) Schematic overview of the density gradient ultracentrifugation (DGUC) protocol used to isolate pEV from expired platelet concentrates (PC). (**B**) Expression of platelet activation marker CD62p in basal and thrombin-stimulated platelets (*n* = 3; two-tailed unpaired *t*-test, *** *p* < 0.001). (**C**) Representative transmission electron microscopy (TEM) micrographs of platelets from the expired PC (activated platelets are indicated by black arrows and resting platelets by red arrows). Scale bars: 1 μm. (**D**) Western blot analysis of specific EV markers (tetraspanins CD63 (glycosylated form) and CD9, and cytosolic protein flotillin-2), platelet-specific marker (CD41), and non-EV markers [apolipoprotein (ApoA1) and argonaute 2 (Ago2)] in pooled fractions and platelet lysate (PLTS). Molecular weight (MW) markers are indicated. (**E**) Representative TEM images of negatively stained 8–9 fractions, enriched in cup-shaped pEV (black arrows). Higher magnification image detailing the morphology of pEV. Scale bars: 1 µm and 200 nm (high magnification). (**F**) Representative size distribution profiles of 8–9 pooled pEV-fractions analyzed using nanoparticle tracking analysis. Size distribution is represented as mean (black continuous line) ± standard deviation (shaded area).

**Figure 2 pharmaceutics-15-00953-f002:**
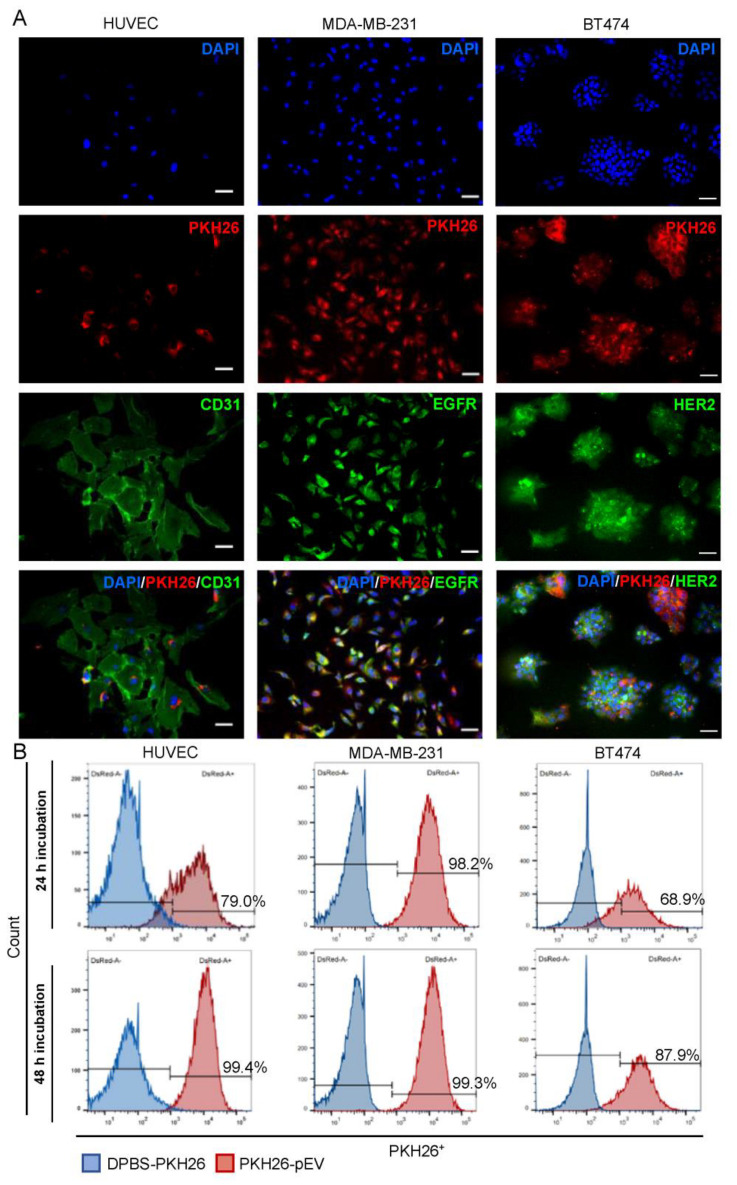
Cellular uptake of PKH26-pEV into HUVEC, MDA-MB-231, and BT474 cells. (**A**) Representative immunofluorescence images of PKH26-pEV uptake. Cells were incubated with PKH26-pEV (Red, at a density of 6000 pEV/cell) for 24 h. HUVEC, MDA-MB-231, and BT474 cells were stained for CD31, EGFR, HER2 (green), respectively, and nuclei (DAPI, blue). Scale bars: 50 μm. (**B**) PKH26-pEV positive cells were quantitatively measured by flow cytometry 24 and 48 h after PKH26-pEV treatment. Blue filled histograms show the negative control population (cells incubated with DPBS-PKH26), and red filled histograms represent the PKH26-pEV population. The *x*-axis represents detection by the DsRed-A filter and the *y*-axis represents pEV counts.

**Figure 3 pharmaceutics-15-00953-f003:**
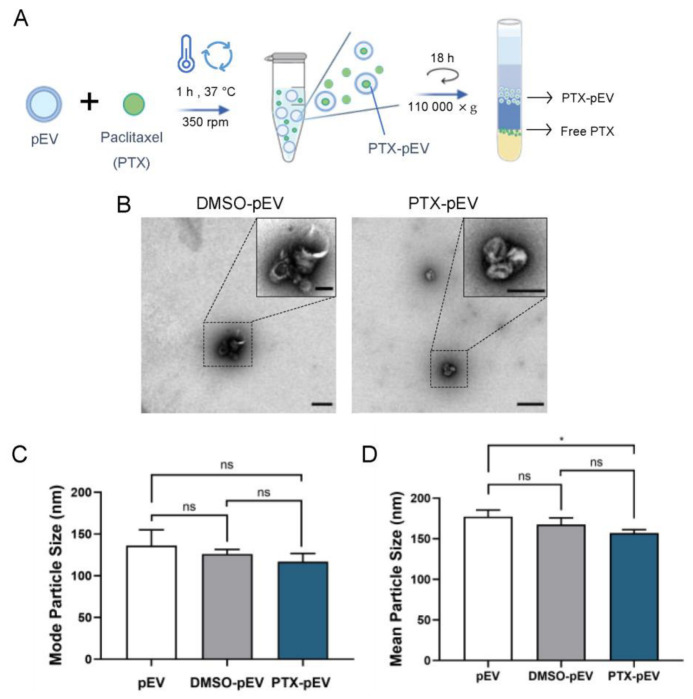
Loading of paclitaxel into pEV by direct incubation and characterization of PTX-pEV. (**A**) Schematic workflow of PTX loading protocol in pEV by direct incubation. (**B**) Representative negative staining TEM images of DMSO-pEV, and PTX-pEV. Scale bars: 500 nm and 200 nm (high magnification). (**C**) Mode and (**D**) mean particle size (nm) of pEV, DMSO-pEV, and PTX-pEV, as measured by NTA (*n* = 3). *n* represents biologically independent replicates. Data are represented as mean ± S.D. One-way ANOVA followed by Tukey’s multiple comparison test, * *p* < 0.05, n.s., not significant.

**Figure 4 pharmaceutics-15-00953-f004:**
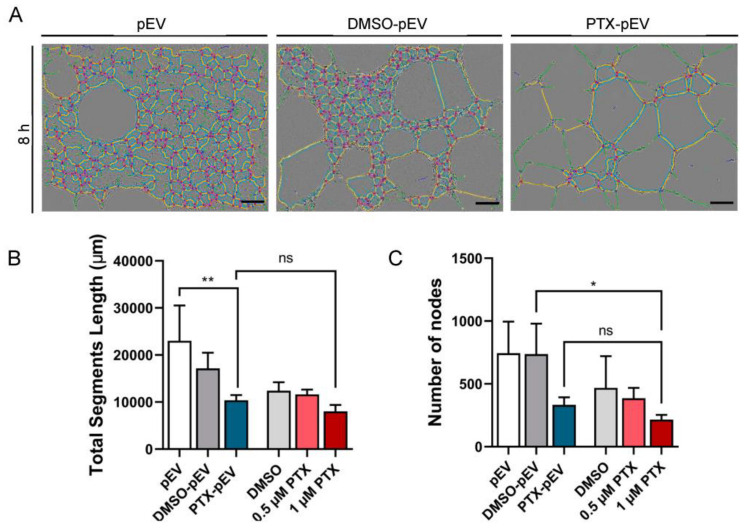
Effect of PTX-pEV on HUVEC angiogenesis. (**A**) Representative images of HUVEC tube formation assay cultured on growth factor-reduced matrigel treated with pEV, DMSO-pEV (treatment vehicle), and PTX-pEV at a density of 6000 pEV/cell. Scale bars: 200 μm. (**B**) Total segments length (µm), and (**C**) number of nodes after 8 h of treatment (*n* = 3). *n* represents biologically independent replicates. Data are represented as mean ± S.D. One-way ANOVA followed by Tukey’s multiple comparison test, * *p* < 0.05, ** *p* < 0.01, n.s., not significant.

**Figure 5 pharmaceutics-15-00953-f005:**
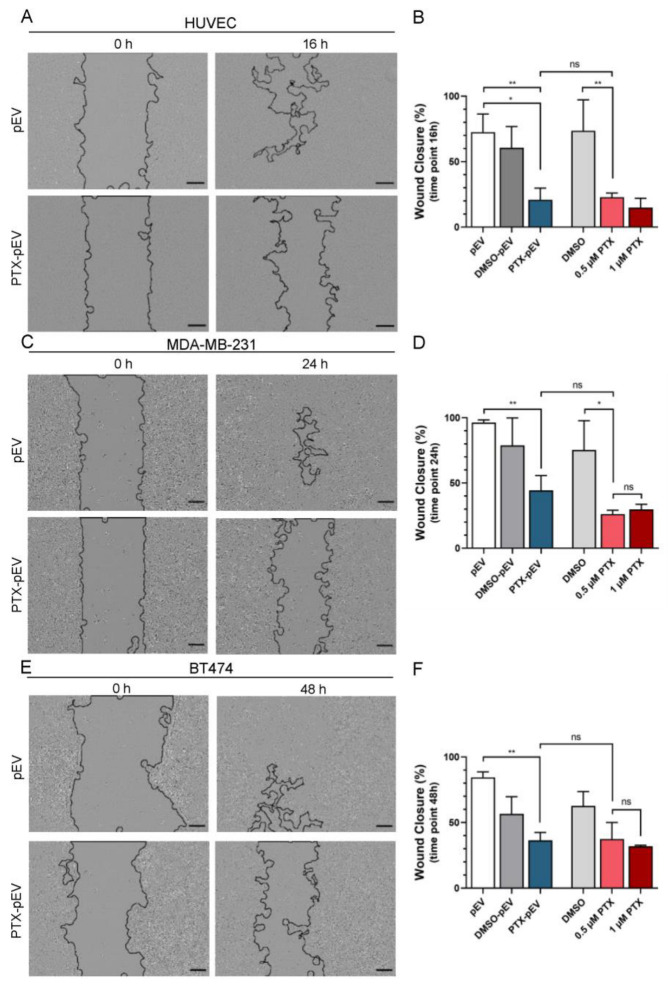
Effects of PTX-pEV on HUVEC, MDA-MB-23, and BT474 cell migration. (**A**,**C**,**E**) Representative images at the initial time (0 h) and wound healing at 16 h for HUVEC, 24 h for MDA-MB-231, and 48 h for BT474 cells incubated PTX-pEV (6000 PTX-pEV/cell). As a control, pEV were used at a density of 6000 pEV/cell. Scale bars: 200 µm. (**B**,**D**,**F**) Quantitative analysis of the wound closure percentage for cells treated with pEV, DMSO-pEV (treatment vehicle), PTX-pEV, free DMSO and free PTX (0.5 μM and 1 µM) at 16 h (**B**), 24 h (**D**), and 48 h (**F**); (*n* = 3). *n* represents biologically independent replicates. Data are represented as mean ± S.D. One-way ANOVA followed by Tukey’s multiple comparison test, * *p* < 0.05, ** *p* < 0.01, n.s. not significant.

**Figure 6 pharmaceutics-15-00953-f006:**
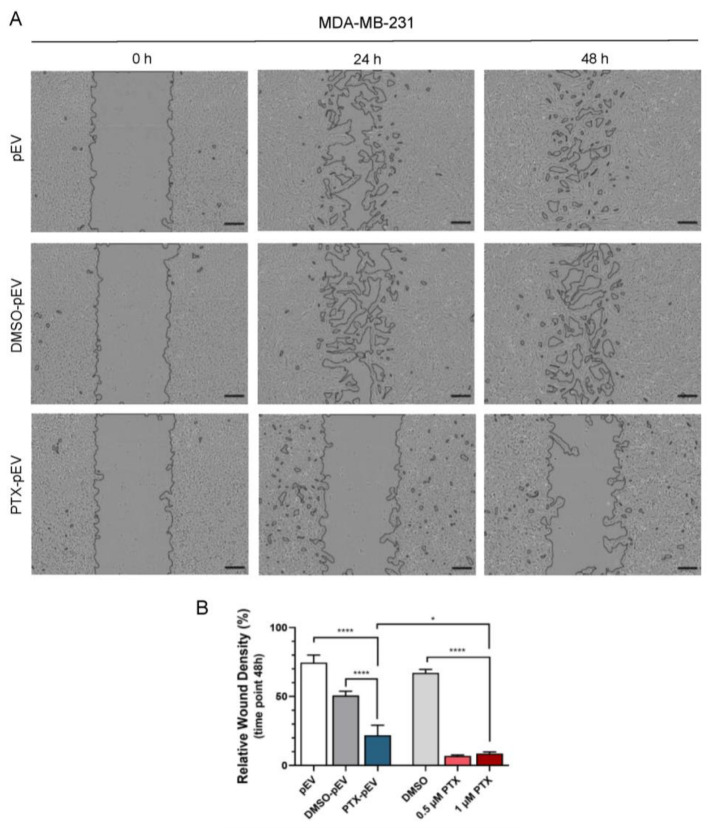
Effect of PTX-pEV on MDA-MB-231 cell invasiveness. (**A**) Representative images at the initial time (0 h) and wound healing at 24 h and 48 h for MDA-MB-231 breast cancer cells incubated with PTX-pEV (6000 PTX-pEV/cell). As controls, pEV and DMSO-pEV (treatment vehicle) were used at a density of 6000 pEV/cell. Scale bars: 200 μm. (**B**) Quantitative analysis of the relative wound density percentage after 48 h of treatment for cells treated with pEV, DMSO-pEV, PTX-pEV, free DMSO, and free PTX (0.5 μM and 1 µM); (*n* = 3). *n* represents biologically independent replicates. Data are represented as mean ± S.D. One-way ANOVA followed by Tukey’s multiple comparison test, * *p* < 0.05, **** *p* < 0.0001.

## Data Availability

We have submitted all relevant data of our experiments to the EV-TRACK knowledgebase (EV-TRACK ID: EV220419) (Van Deun J, et al. EV-TRACK: transparent reporting and centralizing knowledge in extracellular vesicle research. Nature methods. 2017;14(3):228-32.

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
