# Peer review of "Clinically Expired Platelet Concentrates as a Source of Extracellular Vesicles for Targeted Anti-Cancer Drug Delivery"

_pharmaceutics, 2023, doi:10.3390/pharmaceutics15030953_

Round 1

Reviewer 1 Report

I could not accept the idea which is using platelet-derived extracellular vesicles (pEV) as a carrier of the anti-breast cancer drug paclitaxel. There is no explanation of how the pEV can be loaded with paclitaxel, moreover, no way for this kind of targeting.

Author Response

Reply: The authors acknowledge this comment and respect the opinion of Reviewer 1. However, we hope that our response will convince Reviewer 1 to reconsider.

In our study, platelet-derived extracellular vesicles (pEV) were used as carriers of the anti-breast cancer drug paclitaxel for two main reasons:

(1) platelet concentrates are a blood-derived product widely used to treat hematological conditions; however, due to their inherently short shelf-life (5-7 days after platelet collection), approximately 10-20% of donations are discarded on a regular basis. In this context, new alternatives for expired PC are vital to alleviate this financial burden. Indeed, our manuscript proposes a novel application for these products;

(2) large amounts of pEV are released during PC storage (5-7 days), but little research has explored its applicability. Additionally, our rationale for the applicability of pEV as drug delivery vehicles is supported by the well-described interaction between tumor cells and platelet/pEV membrane receptors.

Regarding how pEV can be loaded with paclitaxel (PTX), the protocol employed in our study was direct incubation (as detailed in the Materials and Methods Section, topic 2.4.’ Paclitaxel loading into pEV by direct incubation’). Direct incubation is one of the least time-consuming and simplest drug loading approaches. In this method, the EV and drug are co-incubated, and differing drug concentration gradients contribute for loading. Moreover, since PTX is a hydrophobic drug, it is passively incorporated across the EV membrane, which acts as its carrier.

EV are being extensively studied as advanced drug delivery systems not only at an academic level, but also by several pharmaceutical companies, due to their low toxicity, high biocompatibility, and low immunogenicity. In this context, we believe our concept and our study is highly relevant.

Reviewer 2 Report

Review comments

 Title: Clinically expired platelet concentrates as a source of 2 extracellular vesicles for targeted anti-cancer drug delivery

 Authors: Ana Meliciano, Daniela Salvador, Pedro Mendonça, Ana F. Louro, Margarida Serra

Platelet derived extracellular vesicles offer a new way to deliver drugs. Clinically used platelet concentrates are discarded after 5-7 days, and creates high wastage.  To address this question, in this paper, Meliciano et al. examine how EVs from old platelets bags could be utilized in therapeutics. The manuscript is well-written, and the data provided by the authors is solid. It can be further improved by addressing the following questions:

Minor revisions:

Introduction

Page 2, row 58-61: Although EVs are roughly classified to three separate classes (exosomes, microvesicles and apoptotic bodies), the size is not determining factor. Authors should update this part (for example see https://www.nature.com/articles/s41556-018-0049-8).

Discussion

page 16, row 759-762: Since PC have 30% of plasma, the isolated EVs are not entirely platelet derived.  

Major revisions:

Materials and methods

The number of biological and technical replicates is not seen in the M&M section, please add this information. The information can be found from the figures, but how many platelet concentrate bags were used? What blood type were the donors (bags)?

Page 6, Cellular Uptake of PKH26-labelled pEV: How did the authors came up with the PEV/cell ratio (6000:1)? What was the reason that both cell lines had the same ratio? If dose curve was done, it should be added to the supplementary data file.

Results

Page 9, figure 1. Flow cytometry-based platelet activation: The activation marker CD62P levels were really high, and it does make sense since the platelets are old. For the comparison authors should analyze fresh platelets as well, so the reader have a better grasp of the normal values, and the effect thrombin have in platelets.

Discussion

Page 17 onward. Authors should discuss also the effect of innate PEV properties in addition to their usage for drug delivery. How the cargo of PEVs (growth factors etc.) can affect the recipient cells? Is there any harm? Also is the blood type specific PEVs ok to use in therapeutics, or should we use autologous platelets?

Author Response

1. Page 2, row 58-61: Although EVs are roughly classified to three separate classes (exosomes, microvesicles and apoptotic bodies), the size is not determining factor. Authors should update this part (for example see https://www.nature.com/articles/s41556-018-0049-8).

Reply: We thank Reviewer 2 for the careful reading of our manuscript, critical evaluation, and valuable suggestions. We also acknowledge for his/her useful comments that helped us improve the quality of the manuscript. Please find below a point-by-point answer to all questions raised by Reviewer 2.

Regarding the introduction’s minor revision, we thank the reviewer for suggesting an interesting article by Andries Zijlstra and Dolores Di Vizio. We have modified the indicated sentence (lines 58-60) and added ref. 14 (Andries Zijlstra & Dolores Di Vizio) to provide a more accurate description of EV stratification.

2. page 16, row 759-762: Since PC have 30% of plasma, the isolated EVs are not entirely platelet derived.

Reply: We acknowledge the reviewer’s relevant feedback. The starting material (PC) contained 30% plasma and therefore, the final EV preparation is only enriched in platelet-derived EV, but not entirely composed of these vesicles. We have now clarified this point in our discussion (lines 807-808).

3. The number of biological and technical replicates is not seen in the M&M section, please add this information. The information can be found from the figures, but how many platelet concentrate bags were used? What blood type were the donors (bags)?

Reply: We have now modified this sentence in the Materials and Methods Section, topic 2.15. ‘Statistical Analyses’; lines 390-391: ‘All data are presented as mean ± standard deviation (n=3 biological replicates, except where stated. n represents the number of independent experiments performed – indicated in figure legends). To clarify your question, each bag of platelet concentrate represents one biological replicate that was used to isolate pEV for later application in the functional assays. Therefore, if n=3, the pEV were isolated from 3 different platelet concentrate bags (3 biological replicates). For the donor bags, we received bags for all blood types, and we did not select any specific ABO or RhD blood type for use as the starting material in pEV isolation. However, to elucidate this point, we have now specified in the Materials and Methods Section, topic 2.1. (lines 107-108), that all blood donors were considered for our experiments.

4. Page 6, Cellular Uptake of PKH26-labelled pEV: How did the authors came up with the PEV/cell ratio (6000:1)? What was the reason that both cell lines had the same ratio? If dose curve was done, it should be added to the supplementary data file.

Reply: As functional assays describe a quantifiable effect of a certain dose of EV and can be uniformly implemented and reproducible, we compared the effect of different EV doses on HUVEC by performing wound healing and tube formation assays. Thereafter, a dose of 6000 EV/cell was selected as optimal. The authors acknowledge this issue and have now included the results that support the selection of this EV therapeutic dose, in the Supplementary Materials (Supplementary Figure S4 was added). The Supplementary Figures numbering was updated in the revised version, and a new sentence has been added in the main text, mentioning Supplementary Figure S4, and the reason for selecting a dose of 6000 pEV/cell (lines 484-486). The reason why both cell lines had the same ratio was to enable direct dose comparison. As reported previously by Nguyen et al, ‘If doses cannot be compared, information on EV preparation and application might not be sufficient to estimate the impact of EVs as a therapeutic’ (https://doi.org/10.1002/jev2.12033).  

5. Page 9, figure 1. Flow cytometry-based platelet activation: The activation marker CD62P levels were really high, and it does make sense since the platelets are old. For the comparison authors should analyze fresh platelets as well, so the reader have a better grasp of the normal values, and the effect thrombin have in platelets.

Reply:  In the scope of our project, only expired platelet concentrates (PC) were provided by the Portuguese Institute for Blood and Transplantation (IPST, Lisbon, Portugal). Indeed, due to donor shortage, our authorization only encompassed expired PC, in order to not compromise the supply of PC to patients in need. Therefore, it was not possible to detect the activation marker CD62P in fresh platelets.  Nevertheless, the authors acknowledge this comment, and we have now included a new sentence in the Discussion Section (lines 803-806) to provide a clearer interpretation of our flow cytometry-based platelet activation results. We have now included information about the CD62P expression levels already described in the literature for fresh and old platelets, as well as added ref. 57.

6. Page 17 onward. Authors should discuss also the effect of innate PEV properties in addition to their usage for drug delivery. How the cargo of PEVs (growth factors etc.) can affect the recipient cells? Is there any harm? Also is the blood type specific PEVs ok to use in therapeutics, or should we use autologous platelets?

Reply: We thank reviewer 2 for the suggestion that will help us to improve the quality of our discussion. We have now added to the Discussion Section new insights into the importance of autologous platelet sources for personalized medicine. Indeed, using autologous sourcing, a lower risk of immune rejection has been reported (ref. 58; lines 810-812). We have also added to the Discussion Section a few sentences discussing the importance of considering the effect that pEV intrinsic features (e.g., growth factors, miRNA, cytokines) may have on recipient cells (lines 939-943; ref 10). We stated that despite the several advantages described for pEV (biocompatibility, low immunogenicity, targeting, etc.), if the aim is to develop PTX-pEV into recognized clinical therapeutic entities, some unknowns must be addressed, such as examining how these features might complicate the prediction of its pharmacodynamics.

Reviewer 3 Report

The clinical translation of EVs into a therapeutic platform for smart drug delivery system remains challenging. It is necessary to establish optimized methods for loading desired drugs into EVs and delivering them either to circulation or specific tumor cells.

In this manuscript, the authors described that platelet-derived extracellular vesicles (pEV) can be used as a carrier of the anti-breast cancer drug paclitaxel for cancer treatment. The authors used and compared three isolation methods (DGUC, SEC, DGUC-SEC) to obtain pure qEV. The anti-tumor efficacy of PTX loaded pEV was evaluated in breast cancer cells.

My comments are the following:

1. In Fig 1B, author described the CD62p expression is higher in thrombin-activated platelets than in basal platelets. However, in the graph, the difference seems insignificant.

Plus, to claim pEV isolated from stimulated platelets, it is crucial to show the presence of EV-associated marker such as annexin V in basal and thrombin activated platelets using flow cytometry analysis. To claim your samples are not contaminated by non-EV structures, author need to show more EV negative marker such as Calnexin in figure 1B.

2. In Fig 2, the authors need to explain why CD31, EGFR and HER2 were used.

3. In Fig S5, images of DMSO-pEV labeled with PKH26 should be represented as control.

4. To prove clearly the loading of paclitaxel into pEV, authors need to confirm the presence of CD47 and CD41, which are specific platelet membrane translocation protein in PTX-pEV.

5. To claim the anti-cancer effects of PTX-EV in TNBC cells, the authors need to use more TNBC cell lines and other subtypes of breast cancer cells.  

6. The cytotoxicity of PTX-EV vs PTX alone in TNBC cells should be evaluated. The stability of PTX-EV also needs to be determined.

7. To identify a potential of clinical application of pEV as a drug delivery tool, in vivo tumor targeting ability of pEV or Biodistribution study should be investigated.

8. It is necessary to evaluate the optimal conditions for platelet storage to set their therapeutical capability.

Unlike traditional synthetic therapeutics, natural materials’ carriers such as EV have many advantages including low toxicity, longer cycle life, high drug loading, high biocompatibility and ability to cross biological barriers. Platelets contribute to hemostasis, immune escape, pathogen interaction and tumor metastasis. In that respect, platelet EV study is worth as a new strategy to design delivery systems of drugs.

In conclusion, PTX-pEV had a similar anti-cancer effect to free PTX. It is difficult to say that PTX-pEV has more advantage than PTX in this manuscript. In addition, to use pEV as a drug delivery tool, critical determinants of its phenotype and function are required.

Author Response

My comments are the following:

Reply: We thank Reviewer 3 for the valuable comments on our manuscript. Please find below a point-by-point answer to all questions raised by Reviewer 3.

1. In Fig 1B, author described the CD62p expression is higher in thrombin-activated platelets than in basal platelets. However, in the graph, the difference seems insignificant.

Plus, to claim pEV isolated from stimulated platelets, it is crucial to show the presence of EV-associated marker such as annexin V in basal and thrombin activated platelets using flow cytometry analysis. To claim your samples are not contaminated by non-EV structures, author need to show more EV negative marker such as Calnexin in figure 1B.

Reply: We acknowledge the comments from Reviewer 3. In the flow cytometry analysis (Figure 1B), we aimed to show the expression level of CD62P in platelets to characterize our starting material (PC) and evaluate the platelet functionality by the addition of an agonist, such as thrombin. This assay was not meant to claim that pEV were isolated from stimulated platelets. Regarding the percentage of CD62P expressed in basal platelets (85.9%) and thrombin-activated platelets (96.38%), an unpaired t-statistical test was performed to determine whether there was a significant difference between the two groups. A significant difference was observed between the means of the two populations, and according to the alternative hypothesis (H1) – one of the hypotheses of an unpaired t-test, this difference is unlikely to be caused by sampling error or chance. And as indicated here, the standard deviation was minimal between groups (Basal: n=1 (87.68%); n=2 (86.00%); n=3 (84.15%) and Thrombin: n=1 (96.98%); n=2 (97.03%); n=3 (95.13%)). Nevertheless, in response to your suggestion, we have now modified the appearance of the graph for a clearer interpretation by readers. Regarding the comment on purity, after characterizing the starting material, we performed a complete analysis of EV-associated markers (CD63, CD9, FLOT2, and CD41) and non-EV markers (ApoA1 and Ago2) – please see figure 1D. We consider that our western blot data complies with the minimal requirements reported by the International Society for Extracellular Vesicles (MISEV) guidelines, which recommend characterizing EV preparations for both trans-membrane- (e.g., CD63, CD9), cytosolic- (e.g., FLOT2) and contaminating non-EV proteins (e.g., ApoA1, Ago2). We were able to show the presence of EV-associated markers and the lack of non-EV structures, as revealed by the absence of ApoA1 and Ago2 in fractions 8-9. Despite this, we appreciate the comment of Reviewer 3 regarding the possibility of using additional EV-associated markers (e.g., annexin V) and other EV negative markers (e.g., calnexin), and we have now included a remark in the Discussion Section mentioning the MISEV guidelines (lines 817-818, ref 59 was also added).

2. In Fig 2, the authors need to explain why CD31, EGFR and HER2 were used.

Reply: The authors agree that an explanation for the use of CD31, EGFR and HER2 markers should be included in our manuscript, having this question also been raised by the Academic Editors. These are markers expressed by endothelial (e.g., HUVEC) and breast cancer cells (MDA-MB-231 and BT474), respectively, and were used to stain cells following the PKH26-pEV uptake assay to show EV internalization. According to the literature, CD31 (also known as platelet endothelial cell adhesion molecule) is a specific marker for vascular endothelial cells. EGFR is overexpressed in MDA-MB-231 TNBC cells, and the BT474 cell line highly expresses HER2. We have now explained in the Discussion Section (lines 860-864) the reason for choosing these antibodies. Please note that also three new refs. (67,68, and 69) have been added.

3. In Fig S5, images of DMSO-pEV labeled with PKH26 should be represented as control.

Reply: To confirm PTX entrapment into pEV and their uptake by endothelial cells, we used PTX488 (green) and PKH26-pEV (red), respectively. Therefore, in this experiment, we studied the co-localization of PTX488 (green) and PKH26-pEV (red), aiming to demonstrate the successful trapping of PTX within pEV. We did not consider including DMSO-pEV labeled with PKH26 to be a relevant control. In this experiment, we used PTX488 and PKH26-pEV, therefore we believe that possible controls should be the absence of one of these variables [i.e., a) unlabeled pEV entrapped with PTX488 or b) PKH26-labelled pEV entrapped with PTX488].

4. To prove clearly the loading of paclitaxel into pEV, authors need to confirm the presence of CD47 and CD41, which are specific platelet membrane translocation protein in PTX-pEV.

Reply: The authors agree that the presence of key platelet membrane proteins, such as platelet specific CD41 and immunomodulatory CD47, represent important markers to confirm the origin of EV. The western blot results shown in Fig. 1D, already demonstrates that this aspect was taken into account in our manuscript. Nevertheless, we appreciate the comment of Reviewer 3, since after drug loading, we should have confirmed that the platelet-specific membrane was retained (repeating the western blot for PTX-pEV samples against CD41 and additionally CD47 could also be included). Instead, we performed TEM (morphology) and NTA (size) in EV samples before and after PTX entrapment by direct incubation, and no significant changes were observed in EV. Moreover, no significant changes in EV specific membrane translocation proteins have been described in the literature following drug loading by direct incubation. Nevertheless, we have now included this possible limitation of our work in the Discussion Section (lines 884-886, ref. 22 was also added).

5. To claim the anti-cancer effects of PTX-EV in TNBC cells, the authors need to use more TNBC cell lines and other subtypes of breast cancer cells.  

Reply: We have now added new sentences in the Discussion Section (lines 899-904), stating that due to breast cancer heterogeneity, different TNBC cells may respond differently to treatment, and suggesting that multiple cell lines might help predict the response to drug therapeutics. We understand the reviewer’s concern but we want to highlight that our study is a proof-of-concept on the use of expired PC as valuable source of EV for drug delivery, and not an exhaustive study on the effect of PTX on different breast cancer subtypes. Two different subtypes of breast cancer cell lines were carefully selected, MDA-MB-231 and BT474. MDA-MB-231 (basal origin) is a cell line with high-growth and metastatic potential and has a triple-negative profile (PR-, ER-, HER-). In contrast, BT474 (luminal origin) is a cell line with a slow-growing, weakly invasive, and with a triple-positive profile (PR+, ER+, HER+). We have now included in the Discussion Section (lines 899-904) a more thorough reasoning for choosing these two breast cancer cell lines. We also added refs. 76 and 77.

6. The cytotoxicityof PTX-EVvs PTX alone in TNBC cells should be evaluated. The stability of PTX-EV also needs to be determined. 

Reply: Regarding the cytotoxic effects of PTX-pEV and PTX, no specific assay was performed, since PTX has a cytostatic effect at the concentrations used in our experiments, as described in the literature (Pasquier et al, 2004). For concentrations <10 nmol/L, PTX had a cytostatic effect defined as an inhibition of cell proliferation without apoptosis. In contrast, at concentrations of ≥10 nmol/L, PTX had a cytotoxic effect with the induction of apoptosis.

Concerning the preservation and storage stability of PTX-pEV, we have now included a remark in the Materials and Methods Section (lines 191-192 and lines 299-300), as well as in the Discussion Section (lines 888-898). Briefly, in Materials and Methods, we stated that PTX-pEV and PKH26-pEV were used for functional assays immediately after drug loading or PKH26 labeling, respectively, and if not possible, stored at 4 ËšC for a maximum of one week until further use. In the Discussion Section (lines 888-898), we have added ref. 75 to support the fact that the storage stability of drug-loaded EV (even when using a non-aggressive method, such as direct incubation) should be affected by various storage temperatures. The EV content seems to be more stable at 4 ËšC, than after repeated freeze-thaw cycles. Hence, we based the stability of PTX-pEV and PKH26-pEV on the literature and did not study in detail. We consider that such an evaluation will be more relevant when considering off-the-shelf applications, which is not the case.

7. To identify a potential of clinical application of pEV as a drug delivery tool, in vivo tumor targeting abilityof pEV or Biodistribution study should be investigated.

Reply: We agree that the lack of evaluation of PTX-pEV tumor targeting abilities, biocompatibility, and biodistribution patterns using in vivo studies can be considered a limitation of our study. In the last paragraph of the Discussion Section of the unrevised manuscript, we have mentioned these limitations, and disclosed that including these assays may be a follow-up to this manuscript. We have now elucidated the importance of evaluating these aspects to achieve clinical translational applications of EV (lines 958-959). It is also worth mentioning that, while these studies are highly encouraging in preclinical proof-of-concept models (research using animals to study the potential of therapeutic products), because we consider our manuscript to be a proof-of-concept in discovering a new application for expired PC using PTX-pEV carriers, such applications should be a follow-up study and not to be included in our manuscript.

8. It is necessary to evaluate the optimal conditions for platelet storageto set their therapeutical capability.

Reply: Platelet concentrates were prepared at the Portuguese Institute for Blood and Transplantation (IPST, Lisbon, Portugal), where optimal conditions for platelet storage were fulfilled, following the Portuguese and European regulatory legislation as stated in the ‘Guide to the preparation, use, and quality assurance of blood components’ .The optimal conditions recommended to guarantee platelet viability and haemostatic activity are storage at room temperature (20-24 °C) with continuous agitation and using plastic containers with increased oxygen permeability. Current blood-banking recommendations are that platelet concentrates should not be stored longer than 5-7 days (limited shelf-life), so PC that were not used for clinical transfusion at this time, were provided by the IPST to us to perform the experiments presented in our manuscript. One or two days after their shelf life (when they can no longer be used for transplantation), PC were received and processed in our lab for EV isolation. In the unrevised version of our manuscript, we mentioned in the Material and Methods Section that optimal conditions of PC storage followed the established guidelines, and that after their shelf life for transfusion, were processed for a maximum of 2 days. However, we have now added to the Discussion Section of the revised manuscript a more detailed explanation stating that optimal conditions were used in our study to store platelet concentrates (lines 794-799).

#Comment#

'Unlike traditional synthetic therapeutics, natural materials’ carriers such as EV have many advantages including low toxicity, longer cycle life, high drug loading, high biocompatibility and ability to cross biological barriers. Platelets contribute to hemostasis, immune escape, pathogen interaction and tumor metastasis. In that respect, platelet EV study is worth as a new strategy to design delivery systems of drugs'.

'In conclusion, PTX-pEV had a similar anti-cancer effect to free PTX. It is difficult to say that PTX-pEV has more advantage than PTX in this manuscript. In addition, to use pEV as a drug delivery tool, critical determinants of its phenotype and function are required'.

Reply: We thank the reviewer for this final comment/feedback on our manuscript. Our study provides proof-of-concept on the use of expired PC as a valuable source of EV for drug delivery and we agree that to use pEV as a drug delivery tool, critical determinants of its phenotype and function are required, as stated by Reviewer 3. We thank the reviewer for the valuable suggestions that contributed to improving the quality of our work.    

Reviewer 4 Report

In this contribution, the authors focus on employing expired platelet concentrates as a source of extracellular vesicles for targeted anti-cancer drug delivery. The proposed approach is of high interest to the scientific community working in the EV field. The manuscript is very well prepared, and I appreciate the supplementary information. After a long time, I was happy to review the manuscript, which could be accepted for publication in the submitted form. Due to this, I have minor comments that should be considered as suggestions for improving the article and future work.

1.      Page 3, lines 108-110 (& 130): Once the shelf-life was reached, the PC was centrifuged for 10 min at 1000 × g (5010R centrifuge, Eppendorf) at room temperature, and the supernatants were stored at - 20 °C.

1.1.   Please specify (give a range) of time needed to reach shelf-life under these conditions. How do you determine it?

1.2.   Why do you use sou low g force (1000 g and 2000g in the next step, both for 10 min) to remove platelets? I will suspect that under these conditions, you will have a lot of remaining platelets in the supernatant. Did you check it by flow cytometry (CD41, plt cloud at plt gate)?

The current standard for platelet removal (from plasma) is 2 x 2500 g. Also, there is a solid suggestion for using a single-step centrifugation protocol at 5000 g for 20 min.

Linda G. Rikkert, Frank A. W. Coumans, Chi M. Hau, Leon W. M. M. Terstappen & Rienk Nieuwland (2021) Platelet removal by single-step centrifugation, Platelets, 32:4, 440-443, DOI: 10.1080/09537104.2020.1779924

2.      Page 9, Figure 1E: Do you have any explanation for why you observe two pEV populations? I will be curious if you will observe the same pater if the removal of platelets from the PC is performed at a higher centrifugation force (i.e. if the 202 nm population is not from the remaining platelets – effect of cooling/freezing). Later, you discuss plt activation (p 16, lines 755-762) but in another context.

3.      It will be of interest to include in future (in vivo) studies clinically used nanoparticle albumin-bound PTX as a positive control. Do you expect a change in the pharmacokinetics of PTX in the PTX-pEV formulation? Just speculation.

Formal

4.      Please use uniform style (MDPI guidelines) for the abbreviation “First Author et al.,” e.g., p:17, l:793 vs p:18, l: 839 vs p:18, l: 841 etc. Italic vs normal text + points.

5.      Readability - numerical values

5.1.   Page 10, line 495: An entrapment efficiency of approximately 5.92± 0.01% of EV-bound PTX was observed… This sentence sounds non-consistent: approximately followed by an (unnecessary) exact number. Or you can omit „approximately“ or a shorter number to 5.9% (as later in the text p:17, l:812) or even to „6%“ for better readability.

5.2.   I will simply the paragraph on page 13, lines 634-637, by omitting decimal numbers. I found it disturbing (and unnecessarily detailed) as in the same paragraph % are reported without decimal values as “~X%”.

6.       Supplementary material: I suggest including the first cover page (title, authors, address, etc.) and putting the figures starting from the second page. The current style of the first page looks overcrowded, and you are forced to place the description of Figure S2 on the next page… MDPI supplementary material is a free form, and of course, I will accept your preferences. Please consider it just as a recommendation.

Author Response

In this contribution, the authors focus on employing expired platelet concentrates as a source of extracellular vesicles for targeted anti-cancer drug delivery. The proposed approach is of high interest to the scientific community working in the EV field. The manuscript is very well prepared, and I appreciate the supplementary information. After a long time, I was happy to review the manuscript, which could be accepted for publication in the submitted form. Due to this, I have minor comments that should be considered as suggestions for improving the article and future work.

Reply: We thank reviewer 4 for the positive feedback and valuable suggestions on our study that helped us to improve the quality of the manuscript. Please find below a point-by-point answer to all questions and comments raised by Reviewer 4.

1. Page 3, lines 108-110 (& 130): Once the shelf-life was reached, the PC was centrifuged for 10 min at 1000 × g (5010R centrifuge, Eppendorf) at room temperature, and the supernatants were stored at - 20 °C.

1.1.   Please specify (give a range) of time needed to reach shelf-life under these conditions. How do you determine it?

Reply:  We have now modified the sentence in line 114, as presented in Materials and Methods Section, topic 2.1. ‘Platelet Concentrates Collection, Quality Control and Processing’, to describe more clearly to the reader in what circumstances PC were used. Regarding the time needed to reach PC shelf-life, it has been described by the healthcare system guidelines, that after platelet collection, PC should be stored under optimal conditions for up to 5-7 days (doi: 10.1016/j.transci.2014.08.006, ref. 4). The optimal conditions include: (a) an appropriate storage temperature of 22 ± 2°C; (b) pH values above 6.0; (c) specific plastic materials that can be in contact with PC, and (d) constant agitation to ensure effective gas exchange. Please note that, as already described in lines 105-106 of the unrevised version of our manuscript, compliance with these storage requirements has been ensured by our collaborators and co-author from the Portuguese Institute for Blood and Transplantation (IPBS, Lisbon, Portugal). After shelf-life of PC was reached (5-7 days after platelet collection), questions related with PC safety arise. Therefore, platelet donations are no longer suitable for clinical transfusion. To decrease PC wastage (< 2 days after its expiration date), PC were processed, centrifuged for 10 min at 1000 x g and the supernatants stored at -20 C until pEV separation.

1.2.   Why do you use sou low g force (1000 g and 2000g in the next step, both for 10 min) to remove platelets? I will suspect that under these conditions, you will have a lot of remaining platelets in the supernatant. Did you check it by flow cytometry (CD41, plt cloud at plt gate)?

The current standard for platelet removal (from plasma) is 2 x 2500 g. Also, there is a solid suggestion for using a single-step centrifugation protocol at 5000 g for 20 min.

Linda G. Rikkert, Frank A. W. Coumans, Chi M. Hau, Leon W. M. M. Terstappen & Rienk Nieuwland (2021) Platelet removal by single-step centrifugation, Platelets, 32:4, 440-443, DOI: 10.1080/09537104.2020.1779924

Reply:  We thank the reviewer for raising this critical question. Our reasoning to remove major contaminants (such as platelets) from PC by centrifugation at 1000 x g (10 min), followed by 2000 x g (10 min), and a final filtration through 0.45 µm pore size was because we combined a previous protocol established in our group with literature information. We have added new references (refs. 32 and 33) in section 2.3. ‘pEV Separation from Platelet Concentrates’ to support the adoption of these steps in our pEV separation protocol. Kuravi et al (ref. 32) describe a protocol where platelets were removed at 2000 x g. Although we have not confirmed the total absence of platelets in our supernatant by flow cytometry, we can assume that after filtration through a pore size of 0.45 µm it is unlikely that platelets would remain. In addition, a final step to isolate pEV is density gradient ultracentrifugation, which separates EV from platelet debris based on differences in density. Nevertheless, we acknowledge this comment and the suggestion of new protocols for platelet removal, which in the future may be interesting for comparison with the current method.

2. Page 9, Figure 1F: Do you have any explanation for why you observe two pEV populations? I will be curious if you will observe the same pater if the removal of platelets from the PC is performed at a higher centrifugation force (i.e., if the 202 nm population is not from the remaining platelets – effect of cooling/freezing). Later, you discuss plt activation (p 16, lines 755-762) but in another context.

Reply:  The authors agree that some hypotheses about the observation of two populations of pEV by NTA would improve the quality of the study. We have added a new sentence (lines 839-841) and ref. 62 to support the discussion of the results shown in figure 1F (page 9). As shown in figure 1F, the representative size distribution profile of pEV isolated by DGUC indicated a bimodal distribution ranging from 100 to 300 nm. Particularly, we observed a main peak at 148 nm and a lower peak at 202 nm, evidencing the existence of two major pEV populations: small and large. The presence of two EV subpopulations with different sizes has also been described for other EV preparations from different biological fluids, including conditioned culture media (KırbaÅŸ et al, 2019, ref 62). A possible explanation to observe two pEV peaks is that a very heterogeneous population of pEV is released from activated platelets: a population consisting of large vesicles derived from platelet membrane gemmation and another of small vesicles with endosomal origin.  Another hypothesis is the presence of two pEV populations derived from activated and non-activated platelets. Other studies suggest that EV size distribution profiles are highly dependent on the isolation method – some of which may be more discriminative than others. Regarding the centrifugation of platelets at a higher centrifugation force (i.e., instead of 2000 x g, centrifuge at 2500 x g, as suggested by the reviewer), we do not believe it would change pEV size distribution profile, since previous studies (e.g., Kuravi et al, 2018, ref. 32) have reported that centrifugation at 2000 x g is enough to remove platelets. In addition, the final pEV isolation step is a density gradient ultracentrifugation which separates EV based on density. Platelets and platelet debris are denser than EV and therefore do not migrate upwards on the gradient.

3. It will be of interest to include in future (in vivo) studies clinically used nanoparticle albumin-bound PTX as a positive control. Do you expect a change in the pharmacokinetics of PTX in the PTX-pEV formulation? Just speculation.

Reply: The authors acknowledge the reviewer’s interesting point of view. Albumin-based nanoparticles have been demonstrated to be effective drug delivery systems, owing to their inherent advantages as protein nanocarriers over other nanomaterials (e.g., biocompatibility, biodegradability, or lower cytotoxicity). Due to the versatility and favorable characteristics of protein drug carriers, we agree with the reviewer regarding the inclusion of albumin-bound PTX as positive control, however, we believe this comparison would be more relevant for in vivo biodistribution assays, as follow-up to this study. Regarding the pharmacokinetics of PTX-pEV, studies have shown that following intravenous administration in mice, EV concentration in the bloodstream declines up to 95% within the first five minutes (Morishita et al, 2014; Morishita et al, 2017; Takahashi et, 2013), which is similar to free PTX (Vlerken et al, 2008). Nevertheless, though EV are rapidly removed from circulation, we expected enhanced tumour accumulation of PTX-pEV over administration of free drug, due to the intrinsic tropism of pEV. We believe we already mention some pharmacokinetics advantages of drug carriers functionalized with platelet receptors on the sixth paragraph of the discussion (lines 944-963).

4. Please use uniform style (MDPI guidelines) for the abbreviation “First Author et al.,” e.g., p:17, l:793 vs p:18, l: 839 vs p:18, l: 841 etc. Italic vs normal text + points.

Reply: We have now standardized the style of abbreviations according to MDPI guidelines, which can be seen in track changes (red) in revised version of the manuscript.

5. Readability - numerical values

5.1.  Page 10, line 495: An entrapment efficiency of approximately 5.92± 0.01% of EV-bound PTX was observed… This sentence sounds non-consistent: approximately followed by an (unnecessary) exact number. Or you can omit „approximately“ or a shorter number to 5.9% (as later in the text p:17, l:812) or even to „6%“ for better readability.

5.2.  I will simply the paragraph on page 13, lines 634-637, by omitting decimal numbers. I found it disturbing (and unnecessarily detailed) as in the same paragraph % are reported without decimal values as “~X%”.

Reply: The authors agree that the presence of several decimal numbers could affect the readability of the results by the reader. The suggestions were addressed, as shown in the track changes (red) from the revised version.

6. Supplementary material: I suggest including the first cover page (title, authors, address, etc.) and putting the figures starting from the second page. The current style of the first page looks overcrowded, and you are forced to place the description of Figure S2 on the next page… MDPI supplementary material is a free form, and of course, I will accept your preferences. Please consider it just as a recommendation.

Reply: The authors appreciate the format recommendation and have altered the layout of the first cover page of the Supplementary Materials accordingly.

Reviewer 5 Report

Dear Authors,

Please, find enclosed my review of your work. Overall I find it scientifically sound and experimentally solid. I think that almost anyone working in the field of EVs knows how convenient EV sources are badly needed.

The work would have indeed benefited even of very preliminary in vivo results as, at this stage, the real novelty resides in the EV source you identified and investigated, and the in vitro result represents a promising proof-of-concept. 

Nevertheless, after carefully reading it, I recommend its publishing after some very minor revisions. Find below my advices:

Minor revisions

- Line 438 and figure 1C: using a different color schemes for the arrows (e.g., black and bright red) may help colorblind people.

- Line 458 and TEM figures: please notice that the cup-shaped trait of EVs observed at TEM is largely considered as an artifact of the technique / preparation protocol, rather than a real feature of EVs.

- Line 490: I think "exploiting" was the intended word, not "exploring".

- Lines 776-779: the reason why you chose DGUC over SEC and DGUC-SEC should be better explained in results section as well, as I see you just quickly mention it in line 405

- Lines 849 -861: maybe this sentence is more suited for the "conclusions" sections

Other clarifications/curiosities:

- I'm just curious, not contentious here.  The platelet concentrates here used have been marked as "expired". Do the authors expect this "condition" of the EV source to hamper a possible translation of engineered pEVs into clinics (from practical or ethical point of view)? In other words: aren't EVs from an "expired" source to be considered "expired", too?

- Do you tried to make a formulation of PTX-engineered liposomes to see how a syntethic carrier performs in vitro against natural EVs?

- Do you have an idea of the therapeutic dose of Paclitaxel normally used to treat breast cancer? Is there any correlation with the Paclitaxel doses you tested in vitro?

Best regards

Author Response

Dear Authors,

Please, find enclosed my review of your work. Overall I find it scientifically sound and experimentally solid. I think that almost anyone working in the field of EVs knows how convenient EV sources are badly needed.

The work would have indeed benefited even of very preliminary in vivo results as, at this stage, the real novelty resides in the EV source you identified and investigated, and the in vitro result represents a promising proof-of-concept. 

Nevertheless, after carefully reading it, I recommend its publishing after some very minor revisions. Find below my advices:

Reply: We thank reviewer 5 for the valuable comments on our manuscript. Please find below a point-by-point answer to all questions/comments raised by Reviewer 5.

1. Line 438 and figure 1C: using a different color schemes for the arrows (e.g., black and bright red) may help colorblind people.

Reply: We appreciate the suggestion and now we have altered the blue arrow to a bright red arrow.

2. Line 458 and TEM figures: please notice that the cup-shaped trait of EVs observed at TEM is largely considered as an artifact of the technique / preparation protocol, rather than a real feature of EVs.

Reply: We acknowledge the comment of Reviewer 5 and modified the sentence (lines 474-475), to a more corrected one, being in consideration that cup-shaped is the typical EV’s morphology assessed through negative staining. We have also included in the Discussion Section (lines 820-836) a clarification regarding that cup-shape is a form of artefact that can occur in the preparation protocol of TEM technique, namely during the drying process. Also, we suggested that Cryo-EM is a powerful method that allows the visualization of the natural structure of EV (a more spherical structure). We have added ref. 60 to support our sentences.

3. Line 490: I think "exploiting" was the intended word, not "exploring".

Reply: The authors agree with the correction, and in the revised version of the manuscript the word ‘exploring’ has been replaced by the word ‘exploiting’ (line 510, track change – red).

4. Lines 776-779: the reason why you chose DGUC over SEC and DGUC-SEC should be better explained in results section as well, as I see you just quickly mention it in line 405

Reply: We appreciate the correction and understand this statement, but since the data from the comparison of the three pEV isolation methods appears only in the Supplementary Materials, we considered to quickly indicate why DGUC was chosen (in the Results Section). Though, we decided to better explain and justify why the DGUC was selected as the optimal separation protocol (lines 422-423).

5. Lines 849 -861: maybe this sentence is more suited for the "conclusions" sections

Reply: We believe that the last paragraph of Discussion Section (lines 944-963) describes a limitation of our study and highlights the added benefit of evaluating the PTX-pEV behavior in vivo, in terms of tropism, circulation time or dosing. As pointed out by the Reviewer 5, ‘The work would have indeed benefited even of very preliminary in vivo results …. the in vitro result represents a promising proof-of-concept’.

6. I'm just curious, not contentious here.  The platelet concentrates here used have been marked as "expired". Do the authors expect this "condition" of the EV source to hamper a possible translation of engineered pEVs into clinics (from practical or ethical point of view)? In other words: aren't EVs from an "expired" source to be considered "expired", too?

Reply: Despite the storage of PC under optimal conditions and the use of pathogen inactivation technologies, its clinical application for transfusion is limited by its short shelf life of only 5 to 7 days (expiration date). After this date, concerns about the safety of PC arise, particularly due to storage conditions that favor the risk of microbial contamination (e.g., temperature of 22 ± 2 C; pH above 6.0; constant agitation). Although PC are no longer available for transfusion purposes, platelets and their released extracellular vesicles remain functional. In fact, several studies have already demonstrated that expired PC can be used for cell culture supplements and regenerative medicine applications. Human platelet lysates produced from PC have been used as a viable and safe alternative to the gold standard fetal bovine serum (FBS) supplementation; expired platelet transfusion units decrease the risk of pathogen contamination and immune reactions associated with FBS. Ongoing preclinical studies have demonstrated the application of expired platelet lysates to treat corneal diseases or neurological disorders. In summary, although PC are considered expired for transfusion purposes, it is due to risk concerns and not to a decline in functionality. Therefore, other bioactive products can be extracted from expired PC, including EV, which in turn will be subjected to different quality control procedures to guarantee their safety.

7. Do you tried to make a formulation of PTX-engineered liposomes to see how a synthetic carrier performs in vitroagainst natural EVs?

Reply: Several advantages of EV as drug delivery systems over synthetic nanocarriers (e.g., liposomes) have been suggested, including their natural targeting ability, non-immunogenic properties, and low toxicity. EV are assembled from a varied mixture of membrane proteins, some of which contribute to tissue targeting and minimal non-specific interactions. In addition, the clinical application of liposomes has encountered considerable biological barriers such as rapid clearance from the bloodstream, off-target accumulation in clearance organs, and activation of the innate immune response. Therefore, we believe that comparing the behavior of our delivery formulation (PTX-pEV) with PTX-engineered liposomes would be more interesting to investigate in vivo; thus, it was not considered in this proof-of-concept study.

8. Do you have an idea of the therapeutic dose of Paclitaxel normally used to treat breast cancer? Is there any correlation with the Paclitaxel doses you tested in vitro?

Best regards

Reply: According to the literature, PTX is approved for the treatment of node-positive breast cancer. The recommended dose is 175 mg/m2 administered intravenously over 3 hours on an every-3-week regimen (doi:10.2174/2211738507666190122111224). For an average adult (1.70 m, 60 kg), that dose corresponds to approximately 300 mg of PTX. Therefore, the concentrations that we are using are well below the therapeutic doses of PTX used in vivo to treat breast cancer. Nevertheless, these are within the range of PTX doses employed in in vitro studies. Indeed, it is not possible to mimic the same concentrations of drug that are used in vivo in in vitro studies, since these are two completely distinct systems.

Round 2

Reviewer 3 Report

Authors have addressed most of the comments.  In terms of demonstrating a proof-of-concept using expired PC as a source of EV for drug delivery, the manuscript is acceptable. Therefore, I do not envision any issue to publish this article.